# Novel Tough and Transparent Ultra-Extensible Nanocomposite Elastomers Based on Poly(2-methoxyethylacrylate) and Their Switching between Plasto-Elasticity and Viscoelasticity

**DOI:** 10.3390/polym13234254

**Published:** 2021-12-04

**Authors:** Katarzyna Byś, Beata Strachota, Adam Strachota, Ewa Pavlova, Miloš Steinhart, Beata Mossety-Leszczak, Weronika Zając

**Affiliations:** 1Institute of Macromolecular Chemistry, Czech Academy of Sciences, Heyrovskeho nam. 2, CZ-162 06 Praha, Czech Republic; bys@imc.cas.cz (K.B.); beata@imc.cas.cz (B.S.); pavlova@imc.cas.cz (E.P.); steinhart@imc.cas.cz (M.S.); 2Department of Industrial and Materials Chemistry, Faculty of Chemistry, Rzeszow University of Technology, al. Powstancow Warszawy 6, 35-959 Rzeszow, Poland; mossety@prz.edu.pl; 3Doctoral School of Engineering and Technical Sciences at the Rzeszow University of Technology, al. Powstancow Warszawy 12, 35-959 Rzeszow, Poland; d525@stud.prz.edu.pl

**Keywords:** nanocomposites, transparent, physical networks, elastomers, self-assembly, poly(methoxyethyl acrylate), silica, clay, tough elastomers

## Abstract

Novel stiff, tough, highly transparent and ultra-extensible self-assembled nanocomposite elastomers based on poly(2-methoxyethylacrylate) (polyMEA) were synthesized. The materials are physically crosslinked by small in-situ-formed silica nanospheres, sized 3–5 nm, which proved to be a very efficient macro-crosslinker in the self-assembled network architecture. Very high values of yield stress (2.3 MPa), tensile strength (3.0 MPa), and modulus (typically 10 MPa), were achieved in combination with ultra-extensibility: the stiffest sample was breaking at 1610% of elongation. Related nanocomposites doubly filled with nano-silica and clay nano-platelets were also prepared, which displayed interesting synergy effects of the fillers at some compositions. All the nanocomposites exhibit ‘plasto-elastic’ tensile behaviour in the ‘as prepared’ state: they display considerable energy absorption (and also ‘necking’ like plastics), but at the same time a large but not complete (50%) retraction of deformation. However, after the first large tensile deformation, the materials irreversibly switch to ‘real elastomeric’ tensile behaviour (with some creep). The initial ‘plasto-elastic’ stretching thus causes an internal rearrangement. The studied materials, which additionally are valuable due to their high transparency, could be of application interest as advanced structural materials in soft robotics, in implant technology, or in regenerative medicine. The presented study focuses on structure-property relationships, and on their effects on physical properties, especially on the complex tensile, elastic and viscoelastic behaviour of the polyMEA nanocomposites.

## 1. Introduction

This work is dedicated to the synthesis of novel ultra-extensible tough and strong solvent-free nanocomposite elastomers (xerogels), which were found to display an unusual plasto-elastic and elasto-plastic behaviour. The studied materials are of potential interest for biomedical technologies, as well as for robotics and soft robotics, due to their exceptional mechanical properties, and due to their bio-compatibility.

### 1.1. Ultra-Extensible Elastomers and Hydrogels

The mechanical properties of typical commercial elastomers like natural or synthetic rubber include moduli in the MPa range, as well as extensibilities from 100 to 600%, exceptionally larger [1]. Relatively recently, however, novel ultra-stretchable elastomers were developed: they include both solvent-free elastomers, as well as gels—usually hydrogels. Among these materials, the highly extensible (hydro-)gels are the most investigated sub-group: In spite of partial uncoiling of the elastic chains due to swelling and hence to theoretically somewhat reduced extensibility, they were described earlier (first: Haraguchi-type gels in [2]) and more often in the literature, and they also achieved higher elongations at break than their solvent-free counterparts. They display sophisticated architectures, especially if they possess high moduli (in the MPa range, see [2,3]).

### 1.2. Nanoparticles and Ultra-Extensible Elastomers

The architectures of the ultra extensible (hydro)gels always contain nano-objects of different shape and composition, form which numerous very long polymer chains extend, thus forming brush-like structure segments. The ‘hair’ (long chains) of neighbouring ‘brushes’ are then mutually connected via entanglements, and also by some permanent crosslinks (covalent bonds, trapped entanglements, strong adsorption of chain segments), thus forming the ultra-extensible network. Similar structural patterns are also found in the solvent-free ultra-elastomers discussed further below. The mentioned nanocomposite structures can be divided into organic-organic ones [4,5] (rather rare), and organic-inorganic ones [2,3,6,7,8,9,10,11,12,13,14,15] (typical).

**Inorganic nanofillers** in general can greatly improve the mechanical properties of a given polymer matrix, also in the swollen state, via interface interactions, for which their high specific surface is very advantageous [16]. If all nanofiller dimensions are sufficiently small, optical transparency additionally can be preserved [6,17], while specific chemical [18,19,20,21,22,23,24,25,26,27], optical [28,29], electrical [30,31], magnetic [32,33], or gas barrier [34,35,36] properties can be lent to the matrix. Additionally, in the hyper-elastic gels studied in this work, the inorganic filler plays the role of a key structural unit (multi-functional crosslink) in the complex architecture, so that the heterogeneity on the nano-scale does not reduce the extensibility as would be expected in simple materials, but conversely increases it greatly (architecture effect). The inorganic nanofillers used in the ultra-extensible gels are either particulate [6,7,8], or more often of nano-platelet-type [2,3,9,10,11,12,13,14,15]. Among the latter, graphene oxide [9] formally could be counted to the organic nanofillers. Hyperelastic gels were prepared also using other swelling medium than water, for example with the much less volatile ethylene glycol [8]. Record values of elongation at break for ultra-extensible gels are in the range of twelve thousand to fifteen thousand % (see [5,6] respectively), which means 120 to 150 times the original length.

Nano-gels, structured nano-droplets, as well as core-shell nanoparticles are attractive nanomaterials related to the above-discussed nanocomposite gels. Their structure often is closely similar to the basic structural sub-unit of ultra-extensible nanocomposites, namely to the ‘brush nanoparticle’. The ‘nano-gels’ and similar materials are of interest especially for medical and biomedical applications, often as smart drug-release systems. Examples include polyacrylate copolymer nano-gel particles serving as drug vehicle in cancer therapy [37], pH-sensitive drug-releasing micelles [38], or self-nanoemulsifying drug delivery systems (SNEDDS—nano-droplets) [39]. Also the opposite function of nano-gels was explored, namely the selective removal of specific molecules (e.g., dyes) from the surrounding solution [40]. Structured nanoparticulate materials further offer application as advanced dermal fillers in cosmetology and medicine [41]: Such systems are often based on hyaluronic acid as the main polymer component [42,43]. In the dermal filler applications, the association of the brush nanoparticles (albeit less strong than in the elastomers studied by the authors) also has important mechanical and viscoelastic effects (see e.g., nano-gel with combined anti-cancer and mechanical function [44]).

### 1.3. Solvent-Free Hyper-Elastomers

A marked disadvantage of the hyper-elastic (hydro)gels is their solvent content and hence their tendency to change via swelling/deswelling or drying. **Ultra-extensible solvent-free elastomers** are more difficult to obtain than their hydrogel counterparts, however, and are hence **much less studied in the literature**. Two different groups of such materials can be recognized: super-soft elastomers, as well as hyper-elastic rubbers (with moduli in the MPa range).

**Super-soft solvent-free elastomers** (SSSFE) with moduli similar to soft hydrogels (5000 down to 500 Pa), are based on long linear bottlebrush structures crosslinked either by entanglements, or by covalent bonds, see e.g., [45]. In such materials, the highly flexible side-chains take the same role, which the solvent molecules play in hydrogels.

Concerning the stiffer materials, the first reported **ultra-elastic rubbers** [46] were based on a solvent-free system structurally similar to the nano-platelet-crosslinked hydrogels mentioned further above, namely to polyacrylamide/clay (‘Haraguchi gels’ [2]). However, the solvent-free elastomer was based on the hydrophobic and highly flexible poly(2-methoxyethyl acrylate) (“polyMEA”, glass transition near −20 °C), and contained a higher fraction of elastic chains than its hydrogel analogues. The additional chains serve as a replacement of the solvent contained in hydrogel counterparts (analogy to SSSFE), but at the same time they additionally have an elastic function, also via entanglements. Achieved were extensibilities between 1000 and 3000% [46]. **PolyMEA/clay** was found to display bio-compatibility and has a potential for medicine-related applications like tissue engineering [47,48,49]. In the present work, polyMEA nanocomposite elastomers with new filler architectures and new filler materials were explored.

Another rare example of ‘rubber-like’ super-extensible elastomers are the **triblock copolymers polystyrene-poly(butyl acrylate)-polystyrene** [50], in which self-assembly (nano-phase-separation) leads to a morphology somewhat similar to the above-discussed polyMEA/clay system. Weakly crosslinking (hydrogen bridging) units, present as ‘dopant’ in the elastic chains, were demonstrated to greatly increase the toughness.

Genelally, the combination of dynamic (‘soft’) and of strong crosslinks between elastic chains seems to be a key feature in all the discussed tough ultra-extensible materials.

### 1.4. Application Potential of the Studied Materials

The ultra-extensible elastomers studied in this work offer a considerable application potential in the role of advanced structural materials. In view of the bio-compatibility of polyMEA [47,48,49] these applications can be not only technical, but also bio-medical. Technical applications include soft robotics, while in the biomedical ones, the materials could play a role in the regenerative medicine, for example as a strong structural material in implants [51], but also in more elastic applications like artificial ligament or tendon (analogy to [52]), or like cartilage (analogy to [53]). Related highly-inorganic-filled materials could also be of interest as a tough artificial bone material, in analogy to bone nano-cement based on poly(methyl methacrylate) [54]. Additionally, the same materials could play analogous bio-inspired roles in robotic applications. Related to the mentioned implant or artificial-tissue-applications is the potential of polyMEA as scaffold in tissue engineering [55,56]. The high transparency of the studied polyMEA nanocomposites is a further valuable property, both for some biomedical (e.g., sensors), as well as for technical applications. Finally, highly modified derivatives of the studied nanocomposites could be the basis of nano-gel particles, whose applications as drug-release systems, dermal implants etc. were discussed further above (analogy to [37,38,39,40,41,42,43,44]).

### 1.5. Authors’ Previous Studies of Elastic Nanocomposites

In their earlier work, the authors studied stimuli-responsive nanocomposite poly(N-isopropylacrylamide) (PNIPAm) hydrogels, which were not ultra-extensible, and where the nanofiller played the role of a mechanical reinforcement, which additionally supported the stability of superporous morphologies during deswelling and re-swelling. Particulate fillers like nano-silica [31,57,58,59,60] and nano-TiO_2_ [61] were employed. Especially nano-silica was observed to form efficient hydrogen-bond-crosslinks with polyacrylamide chains. In more recent work, the authors turned their attention to the ultra-extensible PNIPAm/clay nanocomposite hydrogels and to their modification [62,63,64,65]. Among other results, they were able to fine-tune in a very wide range (between 0.5 and 5 MDa, via initiation conditions) the length of elastic chains in these sophisticated nanocomposites, thus greatly improving their elongation at break without compromising the modulus (role of entanglements). In a most recent work [66], the authors were able to dramatically improve the extensibility and thus also the toughness (without compromising the modulus) of the solvent-free ultra-extensible polyMEA/clay nanocomposite elastomer, using the method of growing very long elastic chains. This material, similarly like some PNIPAm/clay hydrogels [11,12] additionally displayed self-healing of disrupted samples at some compositions [66], as well as self-recovery of internal mechanical damage.

### 1.6. Nanocomposites Comparable with the Studied Ones

PolyMEA/silica nanocomposites, or structurally similar systems were not studied in the literature until very recently (2020–2021), when Asai, Takeoka and co-workers published first works about such types of materials: Their first work was about laser-cure-3D-printing of a blood-compatible weakly divinyl-crosslinked polyMEA submicro-composite filled with highly regular commercial silica spheres sized 110 nm [51]. In spite of the large size of the filler, the elastomers displayed good tensile properties, which interestingly improved (including elongation at break) with rising filler content. In a subsequent study [67], the same authors embedded the same silica spheres into poly[di(ethylene glycol)methyl ether methacrylate] matrix (PMEO2MA) without any chemical crosslinker. PMEO2MA is a brush polymer structurally related to polyMEA, but with much larger side-chains. The new nanocomposite displayed improved elongation at break values (up to 670% in [68]), which in a wide range improved with increasing filler amount, except at highest silica loadings. At high silica contents (highly regular arrangement of the spheres), full transparency was achieved in the cornea-inspired material. In [68], also the role of the interfacial layer and the damping properties of PMEO2MA-silica were investigated. The same group finally also prepared a three-component composite elastomer with PMEO2MA matrix, multilayer graphene (25 µm wide, 6–8 nm thick) as 2D microfiller, and the 110 nm silica spheres [69]. Interestingly, the sub-micrometre silica spheres greatly improved the dispersibility of the graphene filler during the solvent-free synthesis. In all the above works, the most interesting products were obtained with ca. 40 vol.% (ca. 57 wt.%) of silica, the highest extensibility was at 670%, while the highest moduli and yield stresses were obtained in materials with reduced extensibility. The polymerizations were carried out in bulk (no solvent) using either photo-initiation, or heat-induced (azo-initiator) radical one.

**In contrast to above, in the presented work**, usually a different, albeit partly similar monomer (MEA) is used. More importantly, the polymerization is carried out in relatively diluted solution, using redox-initiated radical polymerization (not dependent on outside stimuli), which was fine-tuned previously by the authors to yield ultra-long polymer chains (multi-million masses). Even more importantly, much smaller silica nanoparticles are used as filler in our present work, which form in-situ during the polymerization. While being less regular, these particles are 10^4^ times lighter and possess a 500 times larger specific surface, than the ones discussed above. Moreover, their large surface is in the ‘natural state’ with no adsorbed stabilizer: it ends with Si-OH groups and is hence readier to interact via hydrogen bridges. Also, the self-assembly in solution likely contributes to the highly different tensile behaviour, as well as to the high modulus (at low strain) and toughness of our new polyMEA/silica nanocomposites. Additionally, due to the small nanofiller size, the properties of our products achieve a ‘maximum’ at a much lower filler loading, namely at 8 vol.% (15 wt.%).

### 1.7. Aim of the Present Work

In the presented work, the aim was to explore novel solvent-free polyMEA nanocomposite elastomers containing a new filler, namely in-situ-formed fine-grained nano-silica as macro-crosslinker, in view of previous experience with its very strong hydrogen bridging efficiency in hydrogels. The expectation was to achieve high moduli, high strength and high toughness, combined with ultra-extensibility, which all would be of interest for the further-above discussed attractive technical and biomedical applications.

In comparison to the few poly(acrylate ester)/silica systems known from literature, the filler in the presently studied materials was expected to have a much stronger effect, due to its much smaller size and thus much higher specific surface (the latter additionally is stabilizer-free).

As an further goal, the exploration of doubly filled polyMEA nanocomposites containing both clay nano-platelets and in-situ-silica was also planned, in order to assess eventual hierarchical- and synergy effects of the combined fillers.

## 2. Experimental

### 2.1. Materials

2-Methoxyethyl acrylate (abbreviation: MEA; Product Nr.: M2282, reagent grade, purity > 98%) was obtained from Tokyo Chemical Industry Co., Ltd. (short name TCI, Tokyo, Japan) and used as received without further purification. Ammonium persulfate (abbreviation: APS; Product Nr.: 215589, reagent grade, purity 98%), N,N,N′,N′-tetramethylethylenediamine (abbreviation: TEMED; Product Nr.: 411019, grade: “purified by redistillation” by the manufacturer, purity ≥ 99.5%) and tetramethoxysilane (abbreviation: TMOS; Product Nr.: 218472, reagent grade, purity 98%) were purchased from Sigma-Aldrich (Burlington, MA, USA) and used as received without further purification. The synthetic hectorite clay, “Laponite RDS” (chemical composition: Na_0.7_[(Si_8_Mg_5.5_Li_0.3_)O_20_(OH)_4_]; Product code: “Laponite RDS”, laboratory grade, purity 90%: remaining 10% = adsorbed water), consisting of approximately circular platelets (diameter ~30 nm, thickness ~1 nm) and modified with pyrophosphate ions (P_2_O_7_)^4−^ (as dispersion-enhancing agent), was friendly donated by BYK Additives & Instruments (Wesel, Germany). The 10% of water present in “RDS” were taken into account in the calculation of the amount of this clay, which was needed for a given synthesis.

### 2.2. Nanocomposites’ Preparation

PolyMEA/silica and polyMEA/clay/silica nanocomposite elastomers were prepared by in-situ free-radical polymerization of MEA in water, carried out simultaneously with hydrolysis/condensation of TMOS. The neat polyMEA matrix was also prepared as a reference material. If the clay nanoplatelets (Laponite RDS) had to be incorporated, a homogenous aqueous dispersion of RDS clay was employed as reaction medium, instead of pure water. This dispersion was obtained by intensively stirring RDS in water for 24 h (the process of exfoliation was studied in detail in a previous work of the authors [62]).

To prepare the synthesis mixture, the MEA monomer was added either to pure water or to the RDS dispersion and the solution was purged with argon. Next, the precursor of the silica nanoparticles (TMOS) and one of the redox co-initiators, TEMED, was added. Finally, the second co-initiator APS (as a 1% aqueous solution) also was added. After a brief final stirring, the reaction mixture was transferred into an argon-filled mould (internal dimensions: 100 × 50 × 5 mm^3^ form), which consisted of two glass sheets enclosing a rubber spacer. The reaction was left to run at 25 °C for 24 h. The resulting white opaque hydrogel was dried at room temperature for 24 h, and finally at 50 °C under vacuum for 24 h, in order to obtain the final solvent-free elastomer. The amounts of components used to prepare the studied materials are given in Table 1.

The following parameters were kept constant: the concentration of the MEA monomer (and hence of C=C bonds) in the reaction mixture was 0.75 mol/L, the molar ratio of *n*(APS)/*n*(C=C) = 0.00435, and the ratio of *n*(TEMED)/*n*(C=C) = 0.0148. The ratio *n*(Si-from TMOS)/*n*(C=C) was varied in a wide range: 0.113, 0.240, 0.381, and 0.927, corresponding to ca. 5, 10, 15, and 30 wt.% of silica in dry nanocomposites, respectively. Clay concentration was either 0 or 4 wt.% in dry nanocomposite. The abbreviated sample names are listed in Table 1: “R” symbolizes the clay, and “T” silica: e.g., “4R-5T” is a nanocomposite with 4 wt.% of clay and 5 wt.% of silica.

### 2.3. Characterization

#### 2.3.1. Ash Analysis

The content of inorganic fillers in the elastomers was determined by ash analysis. Each sample was placed into a platinum vessel together with the double of its mass of sulfuric acid, and this mixture was slowly pyrolyzed in air. The remaining ash was heated to ca. 1000 °C for 15 min. The pyrolysis with sulfuric acid was repeated once more with the ash. The dry inorganic ash was then weighed, yielding the filler content.

#### 2.3.2. Transmission Electron Microscopy (TEM)—Nanofiller Dispersion

In order to characterize the nanofiller dispersion, transmission electron microscopy (TEM) was employed. Ultrathin slices (approximately 60 nm thick) of the dried hydrogels were cut using the Ultracut UTC ultramicrotome (from Leica, Wetzlar, Germany). The slices were put on supporting Cu grids and observed with the Tecnai G2 Spirit Twin 12 microscope (from FEI, Brno, Czech Republic) in the bright field mode at the acceleration voltage of 120 kV.

#### 2.3.3. Small-Angle X-ray Scattering

The experiments were performed using a pinhole camera (Molecular Metrology SAXS System) attached to a microfocused X-ray beam generator (Osmic MicroMax 002) operating at 45 kV and 0.66 mA (30 W). The camera was equipped with a multiwire, gas-filled area detector with an active area diameter of 20 cm (Gabriel design). Two experimental setups were used to cover the range of the scattering vector q from 0.004 to 1.1 Å−1, where q = (4π/λ)sin θ, λ is the wavelength of the X-rays, and 2θ is the scattering angle. The scattering intensities were put on an absolute scale using a glassy carbon standard.

#### 2.3.4. Tensile Tests

The tensile tests as well as tensile loading−unloading tests of rectangular specimens of NC elastomer were measured using an ARES-G2 (from TA Instruments, New Castle, DE, USA—part of Waters, Milford, MA, USA), at room temperature, with a cross-head speed of 0.25 mm/s. The samples had the following geometry: total specimen length: 15 mm, length between jaws: 4 mm; width: 2 mm; thickness: 1 mm. At least three measurements were carried out for each sample. Presented are the tensile curves closest to the average one.

#### 2.3.5. Hysteresis Tests

The hysteresis tests were carried out using the same setup and the same equipment like the tensile tests, but were carried out as repeated loading/unloading cycles: standard: 2 cycles; exceptionally: 6 cycles. The loading part of each cycle was performed with the same crosshead speed like in the simple tensile tests, until a pre-defined maximum elongation, which was set equal to ca. 50% of the elongation at break of the given material. The unloading part of the cycle followed immediately, with the same crosshead speed. In order to assess eventual regeneration, in some experiments the second cycle was performed after a delay of 30 min (‘resting time’). Standardly, the second cycle followed immediately after the first.

#### 2.3.6. Thermo-Mechanical Properties of Elastomers

Dynamic-mechanical thermal analysis (DMTA) was carried out using an ARES-G2 apparatus (from TA Instruments, New Castle, DE, USA—part of Waters, Milford, MA, USA). The analysed temperature range was from −80 to +100 °C, the heating rate +3 °C min^−1^. The applied oscillatory deformation had the constant frequency of 1 Hz, while the deformation amplitude was varied between 0.01 and 5% (regulated by the “auto-strain” function). The geometry of standard specimens was 25 mm × 4 mm × 1 mm. The temperature dependences of the storage shear modulus (*G*′), of the loss modulus (*G*”) and of the loss factor (tan δ) were recorded.

#### 2.3.7. Differential Scanning Calorimetry (DSC)

The DSC analyses were performed on a DSC822e instrument (from Mettler Toledo, Greifensee, Switzerland), using the STARe version 16.20 System software (Mettler Toledo). The thermal curves were recorded at the heating rate of 10 K/min, under a nitrogen flow of 60 mL/min. Calibration standards were indium and zinc, both supplied by Mettler Toledo.

#### 2.3.8. Thermogravimetric Analysis (TGA)

TGA was performed using a Pyris 1 TGA thermogravimetric analyzer (from Perkin Elmer, Waltham, MA, USA) in a temperature range from 35 to 750 °C (standard range), at the heating rate of 10 °C/min under a constant gas flow of 20.0 mL/min. All analyses were carried out in nitrogen, as well as in air.

## 3. Results and Discussion

### 3.1. Synthesis of the Physically Crosslinked PolyMEA Nanocomposites

In this work, novel highly transparent elastomeric stiff and tough poly(2-methoxyethylacrylate) (polyMEA) nanocomposites with in-situ-formed fine-grained nano-silica were synthesized, as well as their derivatives doubly filled with nano-silica and clay nanoplatelets (see Figure 1, and the discussion of Micro-phase-separation vs. homogenization further below). This was done by means of simultaneous redox-initiated free-radical polymerization of MEA and sol-gel process of TMOS (studied in more detail in [59]) in aqueous solution (Figure 1a). In the doubly filled derivatives, clay (“RDS”) nanoplatelets were additionally dispersed in the reaction medium. During the synthesis, phase-separation was observed (see discussion further below), which is typical for polyMEA formation in water. Soft opaque hydrated samples were first obtained, which after drying become highly transparent (see discussion further below).

**In-situ-nano-silica**, whose surface is covered by non-protected Si-OH groups, was chosen as physical crosslinker, because it should form relatively strong hydrogen bridges to oxygen atoms of the MEA repeat units (see Figure 1b). This should contribute to permanent crosslinks generated by collective adsorption of large chain segments during early stages of polymerization (‘anchoring’ segments of elastic chains), as well as to ‘normal’ random and reversible bridging between repeat units of a polyMEA chain and a neighbouring SiO_2_ particle (see structure details in Appendix A). The ‘soft physical crosslinking’ was expected to provide sacrificial bonds during large deformations, which can dynamically rearrange, dissipating energy and recovering quickly, thus generating an increase in toughness. Additionally, the ‘soft crosslinking’ could raise the modulus at small deformations. The expectations indeed were confirmed by further-below discussed mechanical tests. In contrast to polyMEA/SiO_2_, the strong and soft crosslinking in the already known polyMEA/clay nanocomposites is based on dipole-dipole interactions between partly negatively charged O atoms of polyMEA and the partly positively charged atoms of Si in the clay platelets (see Figure 1c). Such elastomers also were prepared as reference samples, using ca. circular 25-nm-wide “RDS clay” platelets. In the doubly filled nanocomposites, H-bridging between silica particles and clay (see Figure 1d)—if sufficiently strong—could generate a hierarchical filler structure, which indeed was observed by TEM (see further below).

The size of the nano-silica filler employed in this work (as well as in previous studies with polyacrylamide matrix [62,63]) was expected to be similar or comparable like the one of nano-clay used to prepare the literature-known polyMEA/clay networks (e.g., in [46,66]). Hence, a very similar principle of self-assembly and physical crosslinking can be assumed for the novel polyMEA/SiO_2_ nanocomposites, like for polyMEA/clay (see discussion further below and details in Appendix A). The doubly filled derivatives were expected to display a hybrid structure mixed from both simple ones (shown in Appendix A). These assumptions were indeed confirmed by TEM- and X-ray morphology analyses (see further below).

#### 3.1.1. Micro-Phase Separation during Synthesis and Subsequent Homogeneization

The Figure 2 (central row) shows the development of the outward appearance of the aqueous synthesis mixture of the studied nanocomposites, which is connected with their phase structure (also illustrated in Figure 2: top, and bottom row).

The development of phase separation was nearly identical in case of all filler combinations, independently whether the filler was in-situ-nano-silica, nano-clay, or a combination of both (see Figure 2). Even the polymerization of neat MEA displayed a similar behaviour. At first, the reaction mixture is outwardly fully homogeneous (Figure 2, start of middle row): This stage corresponds to the rapid formation (if TMOS is present) of small SiO_2_ nanoparticles via hydrolysis/condensation (sol-gel process) of TMOS—as part of the process shown in Figure 1a further above. This sol-gel reaction was found to be very fast in a previous study, at analogous conditions: less than 1 min (see [59]). Subsequently, MEA monomer, still intact, due to induction time, as well as TMED and APS co-initiators are likely adsorbed on the nano-SiO_2_ particles (Figure 2 top row, start), in analogy to the already known polyMEA/clay nanocomposite formation (see [46,66]). After an induction time of ca. 10 min, polyMEA chain growth starts on the surface of the nanoparticles, initiated by radical generation through collisions of APS and TEMED. The induction time of the MEA polymerization is attributed to the specific behaviour of the used redox initiating system, which yields analogous results like in case of the kinetics of the related poly(N-isopropylacrylamide) (“PNIPAm”)/clay system in [62]. The sudden progress of MEA polymerization coincides with the appearance of turbidity in Figure 2 central row: After initial growth on nanofiller surface (over the adsorbed MEA molecules), the polyMEA chains start to propagate into the surrounding solution (see Figure 2 top), which eventually causes phase-separation. The latter occurs, because in contrast to monomeric MEA, the polyMEA macromolecules are hydrophobic. In the water-rich domains, some filler nanoparticles which are poorly covered, or not covered by polyMEA could accumulate, together with (temporarily) still unreacted MEA.

The mentioned phase-separation manifests itself as increasing turbidity which eventually dominates the reacting mixture (Figure 2 middle row, centre). Finally, after finished polymerization (and previous SiO_2_ generation), the opaque product displays a white colour and a consistence somewhat similar to cottage cheese (Figure 2, middle row, centre). Drying of this material in air (Figure 2, end of middle row) yields elastomers which are fully transparent—if non-filled, or filled only with SiO_2_, and fairly transparent if nano-clay is present (see Figure 3). In case that the nano-silica content was 15 wt.% or more, the reaction mixture shrunk distinctly and symmetrically towards the end of polymerization, thus releasing a considerable amount of water (see Figure 2 middle row, centre). At 15 wt.% of SiO_2_, also the drying behaviour of the wet product was very attractive, and large crack-free monolithic pieces could be easily obtained by simple drying in air. Below 15 wt.% of SiO_2_ (e.g., at 10%, see Appendix A), the polymerization-induced shrinking was not observed, and larger monolithic specimens underwent cracking upon drying. At very high silica contents (30 wt.%, see Figure 3) the nanocomposites become brittle in the dry state, and they also tend to cracking while drying-up (see Figure 3). In case of doubly filled elastomers (with clay + silica), the SiO_2_ content has analogous effects as described above.

#### 3.1.2. Conversion, Gel Fraction and Content of Incorporated Nano-Fillers

The full **conversion of the MEA monomer** during the synthesis was verified by means of infrared spectroscopy (FTIR): The disappearance of the intense C=C stretching peaks of monomeric MEA (at 1620 and 1638 cm^−1^), in the prepared nanocomposites (prior to drying, as well as after drying) indicates full monomer conversion. This is in good agreement with the above-mentioned kinetics investigations by the authors of the chemically related system PNIPAm/clay [62], which contained the same initiator pair and used analogous synthesis conditions (concentration, temperature) like in the present work. The results of the study in [62] suggest that also the polymerization of MEA should be practically finished in ca. 30 min after initiation. The ‘synthesis time’ of 24 h was nevertheless applied, in order to ensure a large reserve of time for the formation of entanglements (additional physical crosslinks), which was found to be somewhat delayed in respect to the acrylate polymerization in [62].

The **real filler content** in the prepared nanocomposites, which was determined via ash analysis, is shown in Table 2. It can be seen that nano-silica (if it is the only filler) is incorporated practically quantitatively, while in case of doubly filled nanocomposites containing also the RDS clay, the filler content is somewhat smaller than expected: this deviation is the greatest in case of the sample 4R-5T.

Both the quantitative monomer conversion and the ash contents suggest, that in case of the synthesis of polyMEA/SiO_2_, the gel fraction is practically 100%, also for the individual components. In case of doubly filled nanocomposites it is somewhat smaller, due to loss of not embedded nanofiller (expulsed together with water during the phase separation stage).

Generally, the results confirm **very efficient crosslinking between nano-SiO_2_ and the polyMEA matrix**. Also, the affinity of nano-SiO_2_ to polyMEA seems to be stronger than that of RDS clay to the same polymer.

#### 3.1.3. Dispersion of the Fillers: TEM

The dispersion of nano-silica or of clay combined with nano-SiO_2_ in the polyMEA matrix was analysed by means of transmission electron microscopy (TEM) (see Figure 4a–f). High-resolution micrographs are shown in Appendix A. In all cases it can be observed, that both nano-SiO_2_ and the clay nano-platelets are evenly distributed on the larger scale.

The primary silica particles were found to be very small and thus poorly visible by TEM; their size eventually was determined to be 3–6 nm by X-ray diffraction (see further below). The diameter of a single clay platelet was observed to be 20–30 nm in Figure 4d, which well corresponds with data from the supplier. The single particles of both fillers are arranged to larger patterns, but are not grown-together or stacked (see Figure 4g–i). However, at low contents of silica, e.g., at 5 wt.% (Figure 4b), a visible nano-phase-segregation of the latter filler in polyMEA can be observed, into silica-rich and silica-deficient domains. The silica-deficient regions have the shape of 400–800-nm-wide ‘bubbles’, separated by up to 200-nm-thick layers which are rich in nano-silica grains. At 15 wt.% of silica (Figure 4c), the distribution of the nano-SiO_2_ grains is much more even also on the fine scale, and there are much fewer and much smaller silica-deficient domains.

In case that **silica and clay are combined** in doubly filled nanocomposites, it can be seen (Figure 4e,f vs. Figure 4b,c) that the general morphology is similar to the one of nanocomposites filled exclusively by silica. Additionally, the clay platelets (or more precisely their groups) are always buried in the SiO_2_-richer domains in the doubly filled nanocomposites. This indicates the mutual affinity of both fillers, as illustrated in Figure 1d (further above). The more detailed structure of the self-assembled elastomeric networks hence can be described by the Figure 4g–i, in view of the morphology images.

An interesting finding was, that in spite of the periodic fluctuations in nanofiller concentration, which were close to the micrometre scale in the extreme case (see Figure 4b,e further above), no decrease in optical clarity was observed for the low-filled polyMEA/silica elastomers (see Figure 3 further above). This is in contrast with the results obtained for the distantly related nanocomposite based on PMEO2MA-silica (mentioned in Introduction: Asai, Takeoka and co-workers: [67,68]), where filler particles were much larger (110 nm) and where high optical clarity was observed only for highly regular distributions of the filler spheres. In case of polyMEA/nano-SiO_2_ studied in this work, the surprising independence of the nanocomposite transparency from fluctuations of filler distribution can be explained by the close match in the refraction indices of filler and matrix: both are given as 1.46 in the literature (amorphous SiO_2_: [70], polyMEA: [71]).

#### 3.1.4. Morphology Observed by X-ray

In order to characterize the morphology of the prepared nanocomposite elastomers in more detail and to obtain some average structural parameters, X-ray scattering analysis (SAXS, WAXS) was performed (see Figure 5).

In all diffractograms, three characteristic amorphous-halo-type reflections of the **polyMEA matrix** are well visible, namely in the WAXS region, at the right in the respective graphs, between scattering vector (*q*) values of 0.4 and 4 Å^−1^. The central broad peak with maximum at 1.5 Å^−1^ corresponds to the ‘normal’ inter-molecular distance in organic materials (0.425 nm, corresponds to 2 θ = 21° with Cu K_α_ source). The peak at 0.7 Å^−1^ (≡ 0.90 nm ≡ 9.8°) can be assigned to a larger intermolecular distance, most likely the distance between polymer chains which are ‘spaced’ by the pendant methoxyethyl carboxylate groups. Finally, the peak at 3.0 Å^−1^ (≡0.209 nm ≡ 43.2°) might be associated with some short intramolecular distance most likely related to the pendant groups: It was namely found to change in intensity if silica loading was increased to high values, e.g., to 15 wt.% (see Figure 5a,b), or after an intense deformation. The pendant groups are involved in H-bridging to nano-SiO_2_ (see Figure 1b).

In the **low-angle (SAXS) region** of the diffractograms (see Figure 5—left part of the respective graphs; or see zoomed and background-subtracted views in Appendix A), at *q* values below 0.4 Å^−1^ (down to 0.006 Å^−1^), some features characteristic of the phase structure of the nanocomposites can be observed: In case of the neat polyMEA matrix at *q* below 0.05 Å^−1^, a straight increase of scattering intensity with decreasing *q* is observed. The slope of the change in intensity is relatively steep (−3.8), which suggests some spherical fluctuations in density—probably a residuum of the phase separation during the synthesis in aqueous solution (prior to final drying). If polyMEA is filled with in-situ SiO_2_, with clay nano-platelets (RDS), or with a combination of both fillers, the steep change in intensity begins already at higher *q* values, namely below 0.4–0.3 Å^−1^.

In case of **nano-SiO_2_-filled samples**, this straight slope in is always overlaid with one or more broad maxima. In the clay-free sample filled with 5 wt.% of SiO_2_, the slope sets on near q = 0.3 Å^−1^, and the overlaid maximum is relatively intense (see Appendix A). The fit for rigid polydisperse spheres and the Guinier analysis suggest an average size of primary particles to be 3.4 nm, while the smallest secondary domains should be sized 9.7 nm. Both results are in good agreement with the TEM image in Figure 4b, where the smallest secondary particle groups can be recognized in the large pattern of the filler. The slope at the highest angles of the SAXS region is −3.8, which confirms spherical primary particles. At the lowest q values (lowest angles) the slope is relatively flat, which corresponds to a lower-dimensional shape of the larger filler patterns (as seen in TEM: Figure 4b). The nanocomposite with 15 wt.% of SiO_2_ (see Figure 5a and SAXS detail in Appendix A), which possesses very attractive mechanical properties, displays a more intense SAXS pattern (than 0R-5T) which begins already below q = 0.3 Å^−1^: there are two overlaid fairly flat maxima, and the course of the scattering intensity in the SAXS region is closer to a straight line. The evaluation by fits (see marks and labels in Appendix A) yields the following results: a larger size of the primary SiO_2_ nanoparticles at 5.4 nm (and their less spherical shape: initial slope at −3.3), as well as the sizes of larger nanofiller groups: 17 and 45 nm. (Larger domains could not be observed by the employed equipment). The straighter and steeper course of the curve in the SAXS region is in good agreement with the observed percolating-3D-domains morphology and with the higher homogeneity on the small scale (TEM: Figure 4c).

In case of the **clay-filled and doubly filled nanocomposites** (see Figure 5b), the following morphological effects in the SAXS-region-patterns were observed: 4 wt.% of clay generate a nearly straight change in intensity, which begins below ca. *q* = 0.3 Å^−1^, and which has a slope of −2.5, thus indicating 2D domains. This is in good agreement with TEM (Figure 4d: isolated groups of approximately parallelly oriented nano-platelets, exfoliated, but relatively close to each other). In case that nano-silica is present together with clay nano-platelets (RDS), similar maxima like in silica-filled samples are overlaid with the (straight) scattering intensity of the platelets, thus yielding information about the grouping of SiO_2_ particles. Fits yield the following results (see labels in detailed views in Appendix A): In case of 5 wt.% of SiO_2_ in addition to 4% of clay, the size of the smallest secondary aggregates appears to be 10 nm, very similar like in the clay-free sample with 5% of silica. Similarly, the nanocomposite with 15 wt.% of SiO_2_ in addition to 4% of clay displays two flat SAXS maxima (like the related clay-free sample), the primary SiO_2_ particle size appears to be 2.9 nm (smaller than in the clay-free analogue) and the sizes of larger groups are 6.3 and 37 nm. In context of these trends, the size of SiO_2_ particles in the system with 5 wt.% of SiO_2_ + 4% of clay can be estimated to also be somewhat smaller than in the clay-free sample (i.e., smaller than 3.4 nm).

### 3.2. Tensile Properties of the polyMEA Nanocomposites

The most attractive property of the studied novel polyMEA nanocomposite elastomers—besides their high transparency—is their ultra-extensibility combined with a high modulus. The latter corresponds to stiff rubbers in case of the most interesting samples (shear modulus 10 MPa, see also further below: Thermo-mechanical properties, or comparison of moduli in Appendix A). Additionally, the nanocomposites were found to display an interesting mechanically-induced transformation from ‘plasto-elastic’ to ‘real elastomeric’ behaviour. The below-discussed investigations focused on determining the basic tensile characteristics in the post-synthesis state, on studying the degree of elasticity and plasticity before and after the mentioned transformation (via cyclic loading tests), on stress-relaxation phenomena (which illustrate the physical nature of crosslinkinng in the studied elastomers), as well as on the effect of ‘mechanical history’ on the tensile behaviour and on the structural features which made possible the transformation ‘plasto-elastic’ → ‘real elastomer’.

In Figure 6, the tensile curves are shown and compared for nanocomposites filled with silica, for doubly filled ones (silica + clay nanoplatelets), and for some reference materials (with clay filler only, or neat polyMEA). Toughness values are compared in Figure 6d–f.

Tensile curves of the **reference samples** are compared in Figure 6a: It can be seen, that the addition of clay nano-platelets to ultra-high-molecular-weight polyMEA (synthesized according the authors’ previous work [66]) tremendously improves the extensibility (at least until 10 wt.% of clay) and thus the toughness of the polyMEA-based material, but not its tensile strength, or its yield stress (at which the plateau in the tensile curve in Figure 6a begins). This behaviour can be attributed to the effects of supramolecular assembly (polyMEA + clay). Nanocomposites with more than 10 wt.% of RDS were not tested, because in [66] it was found, that RDS amounts distinctly higher than 10% generally lead to problematic synthesis and products.

**In-situ-formed and strongly hydrogen-bonding silica nanoparticles as new physical crosslinker** markedly change the tensile curves of the polyMEA nanocomposites (Figure 6b). It can be observed that the nanoparticles tremendously improve the toughness (see comparison in Figure 6e), yield stress, and stress at break in comparison with neat polyMEA, or with polyMEA/clay. Nano-silica additionally also very markedly improves the extensibility of polyMEA (the improvement grows with the loading until 15%; but at 30% the samples become brittle). The improvement in extensibility is smaller than the one achieved by RDS in polyMEA/10% clay, but the clay nanocomposite, on the other hand, is markedly ‘softer’: it displays a lower modulus, yield stress and tensile strength (see discussion of DMTA results further below, and also Figure 6). Also the shape of the tensile curves of silica-filled nanocomposites is visibly different from the ones of polyMEA/clay, especially at higher nano-SiO_2_ amounts: ‘Plastic-like’ curves are observed, with ‘yield peaks’, with plateau regions affected by ‘necking’ phenomena, and with a tensile stiffening region prior to rupture (Figure 6b). This behaviour can be attributed to reversible crosslinking (‘soft crosslinks’: see Appendix A) between polyMEA and nano-SiO_2_ via H-bridges (see Figure 1b further above), which appears to be fairly strong, as also confirmed by results discussed further below.

Tensile curves of **doubly filled (silica + clay) polyMEA elastomers** are shown in Figure 6c (toughness values: in Figure 6f). It can be seen, that combining the fillers leads to a further improvement of yield stress and stress at break. In case of low nano-SiO_2_ content (see polyMEA/4% clay/5% SiO_2_ in Figure 6c), also the extensibility is greatly increased.

At the highest practical nano-silica content (15%), however, the nanocomposite filled exclusively by silica is tougher and more extensible than the doubly-filled one (with 19% of nanofillers in total). This seems to be the result of an ‘over-filling effect’, which can originate in more frequent morphological defects, and hence in easier crack formation and propagation. The doubly filled polyMEA nanocomposites thus are the strongest (yield stress, tensile strength), but their elongations at break are markedly smaller (1038% with 10% of SiO_2_, and 832 with 15% of SiO_2_) than in case of their analogues filled with silica only.

**To sum up**, it can be concluded, that **in-situ-formed SiO_2_ nanoparticles as a novel physical crosslinker** lend the nanocomposites **excellent values of elongation at break**: 1390% with 10%, and 1610% with 15% of SiO_2_. This compares well with hydrogels (see [2,15]) and with solvent-free nanocomposite elastomers (see [46]) of Haraguchi-type, which are physically crosslinked with a larger 2D filler (25-nm-sized nano-clay). The **doubly filled elastomers** achieved the **highest extensibility among all the silica-containing ones**, namely 3 350% in case of 4R-5T. The latter sample also has a high shear modulus (7 MPa), but is ‘soft’ in terms of yield stress and tensile strength (both ca. 0.5 MPa, similar like in samples filled exclusively by clay). Also the **highest yield stress and tensile strength** (3.4 and 3.6 MPa, respectively) were achieved by doubly filled products, namely by 4R-15T.

#### 3.2.1. Elasticity vs. Plasticity: HYSTERESIS Tests in the ‘as Prepared’ State

The **tensile curves of polyMEA nanocomposites** with nano-SiO_2_ but also with clay, which are compared in Figure 6, display a **shape typical for plastic deformation** (like polyethylene), and the SiO_2_-rich ones (with 10–15%) also display the necking effect—even multiple necking, as shown in the inlay photograph in Figure 6.

**However, all the materials compared in**Figure 6 (further above) **are elastomers**, because they display a marked and sometimes even dominant (albeit never quantitative) recovery of tensile deformation (e.g., rapid elastic shrinking of the fragments after the disruption tests). This **‘plasto-elastic’ behaviour** was characterized in more detail by means of repeated hysteresis experiments with large maximal deformations (ca. 50% of elongation at break). Figure 7a,b and Appendix A show the results obtained for the most important samples. Figure 7e–h shows another interesting result, namely the data from the same experiments like in Figure 7a,b, but with evaluation of the second tensile cycle as a new specimen with a new initial geometry (which it acquired by creep while enduring the first cycle).

It can be seen in Figure 7a, that the **neat polyMEA matrix** (a relatively soft material with shear modulus = 0.14 MPa) displays a behaviour, which is closest to full elastic recovery, albeit with a considerable hysteresis (effect of molecular friction): After the stretching, the sample was unloaded, and 77% of the previous deformation (or ¾ of the maximum x-value in Figure 7a) were recovered. The recovery improved to 85% upon start of the subsequent hysteresis cycle, whose immediate recovery upon unloading was 71% of the second applied deformation.

The next best degree of elastic recovery is observed in the nanocomposite **polyMEA/4% clay** (see Figure 7b). Upon unloading from the first stretching, 61% of the initial deformation is recovered, and this value improves to 70% in the moment when the immediately following second loading cycle starts; 57% is recovered immediately upon unloading this latter cycle.

**4R-15T** is more plastic, the recovery is 48, 55, and 43%, respectively, for the first unloading, start of the second cycle, and the second unloading, respectively.

**0R-15T** displays a similar behaviour like 4R-15T, but is even more plastic, with recovery percentage of 43, 52 and 39% in analogous situations like above. However, even in this case, more than 50% of the tensile deformation is elastic, in spite of the ‘plastic shape’ of the tensile curve.

An important circumstance in the above hysteresis tests of ‘plasto-elasticity’ (Figure 7) was, that prior to the first deformation loading, **all the samples were in the ‘as prepared’ (post-synthesis) state**. As will be discussed further below, it was found by more detailed investigations, that an extensive deformation ‘switches’ the ‘as-prepared’ sample from the ‘plasto-elastic’ to a ‘real elastomeric’ state, which is much more elastic and much less plastic than the initial ‘as prepared’ state.

In the same context, another important observation from Figure 7 is, that the area between the curves of the first hysteresis cycle is very large, especially in the nano-filled samples. In case of each second loading cycle, the area between the curves is much smaller. This means, that in the first cycle, the **‘as-prepared’ material acts as a very efficient energy absorber**. The ‘plasto-elastic’ and energy-absorbing behaviour in the ‘as prepared’ state is attributed to **reversible hydrogen bridging between polyMEA and silica** (see Figure 1b further above), and to **its destruction and subsequent ‘reorganization’** during the first hysteresis cycle. The mechanism is discussed in detail in the section “Assumed mechanism of the change in the tensile behaviour” (see further below). The irreversible rearrangement of the nanofiller/matrix contacts (the energy absorption effect) is much more pronounced, if the filler is nano-SiO_2_, and not clay, with which the same effect is considerably smaller in energy units. This apparently is due to the fact, that in contrast to clay, SiO_2_ can bond via H-bridges to polyMEA. The latter difference also is illustrated by the increasing plastic character of the ‘as prepared’ nanocomposites with increasing SiO_2_ content, as noted in the further-above discussion of the hysteresis cycles results in Figure 7b–d.

##### Multi-Cyclic Loading

It was noted in Figure 7b–d, that the nanocomposite products (containing at least one filler) display a distinctly different shape of the second cyclic deformation curve: it is elastomer-like in contrast to the ‘plastic-like’ curve of the first loading, prior to which the sample was in the ‘as prepared state’. An exception was the neat polyMEA matrix, which is not a nanocomposite: the shape, and partly even the stress values of its second deformation cycle are very similar to the first one. In order to further evaluate changes in tensile behaviour caused by previous deformations, repeated cyclic loading tests (6 cycles) were carried out with the nanocomposite 4R-15T (see Appendix A). It was observed that the second cycle, as well as next following ones, yield mutually very similar elastic curves, and that **practically no significant creep occurs between the second and the next cycles**. In contrast to the initial ‘plasto-elastic’ cycle, the hysteresis is relatively slim in all these later cycles.

Another effect, which was evaluated, was an **eventual slow self-recovery:** The Appendix A compares the initial loading cycle with a second loading cycle, which was applied either immediately after the first one, or after a rest time of 30 min. The results in Appendix A indicate that no slow deformation recovery occurs in 4R-15T (or in similar samples). This is illustrated by identical strain values at which resistance (stress) sets-on during the immediately following, or during the delayed second loading cycle. It was also noted (not illustrated in the Appendix A), that the silica-filled (including doubly filled) polyMEA nanocomposites do not display any significant self-healing if disrupted or cut, in contrast to some polyMEA/clay compositions (see [66]). Both second cycles (with rest, or no rest) in Appendix A are fairly similar. However, it can be observed, that in the cycle recorded after 30 min of rest, the (flat) slope in the region of large deformations is less steep in comparison to the second cycle recorded immediately after the initial one. The trend to temporarily somewhat higher moduli, generated by immediately preceding large deformation, was observed systematically in all the tested samples (effect of time needed for ‘soft’ filler-matrix contacts to rearrange after deformation).

In Figure 7 and Appendix A, the local moduli (slopes), as well as the yield-stress-, and stress-at-break values in different regions of the curves of the second and eventual subsequent loading cycles, appear markedly lower than in the initial loading cycle (‘as prepared’ state). This seeming softening, however, can be partly due to creep, which changed the starting geometry of the samples (initial length, as well as cross-section), prior to the second and subsequent deformation cycles.

##### ‘True Properties’ Using Corrected Starting Geometry

The above-mentioned ‘geometry effect’ on data evaluation was eliminated in Figure 7e–h, which shows the same data (first and second deformation cycle) of the same samples which were compared in Figure 7a–d, but with the second cycle **evaluated as a new specimen with a new initial geometry**. In case of such an evaluation, it can be seen, that polyMEA (Figure 7e) displays practically identical curves in subsequent deformation cycles. But in case of the nanocomposite samples (Figure 7f–h), however, their curves in the second cycle still stay strongly different from the first cycle, also in case of eliminated ‘geometry effect’ (Figure 7f–h: blue curves). These elastic-like curves of the second cycles are similar-shaped but much steeper than the second-cycle-curves in Figure 7b–d (red curves), which were evaluated using the original geometry prior to the first cycle. The initial slopes (moduli) of the ‘geometry-corrected’ second curves in Figure 7e–h are always identical (not flatter) like in the first run. This finding was additionally confirmed by measuring moduli in small-deformation oscillatory shear experiments, in different states of the samples (see Appendix A). After the ‘yield region’, the ‘geometry-corrected’ curves of the second cycles in Figure 7e–h continue to show an elastomeric shape, with a relatively steep slope (but flatter than in the small-deformation region), in contrast to the plateau in the curve of the first (‘plasto-elastic’) cycle. All the inorganic-filled samples show this described trend. Multiple cyclic deformations also were evaluated using the respective new initial geometries (see Appendix A), and yield the same conclusion like Appendix A: the material properties display little change, once the nanocomposite has endured the first large deformation cycle, and the plasto-elastic → elastic transformation is permanent (assigned to above-mentioned rearrangement of filler-matrix contacts).

#### 3.2.2. Stress Relaxation in the Nanocomposite Elastomers

The ‘plasto-elastic’ behaviour of the polyMEA nanocomposites in the ‘as-prepared’ state was further elucidated in stress relaxation tests. The results are summarized in Figure 8 (relaxation of 4R-15T) and in the Appendix A, in Appendix A (relaxation of neat polyMEA, 4R-0T, and 0R-15T).

The graph in Figure 8a illustrates the stress build-up upon sample stretching till half the value of elongation at break (red line), as well as the subsequent relaxation of the stress during a period of rest in the stretched state (blue line). The changes in sample length are depicted in Figure 8b. The Figure 8c compares the effects of relaxation (using relative stress values in %, normalized in respect to the initial stress) at three characteristic time points, for the most important nanocomposite samples: 0R-15T (filled only with 15% of silica), 4R-15T (with 4% of RDS clay + 15% of silica), as well as for the reference materials polyMEA and 4R-0T. The comparison in Figure 8c shows, that all the tested samples display a nearly identical kinetics of stress relaxation: The stress relaxes down to 27–20% of its original value in 25 min. Most of this drop (by ca. 50%, or ca. ¾ of the whole relaxation process) occurs in the first minute, and after 5 min of relaxation, the changes are already very small. The nearly identical relaxation kinetics independently of the presence or absence of the nano-sized crosslinker(s) indicates, that the polyMEA chains (their entanglements and their friction) are the main source of the observed kinetics behaviour.

#### 3.2.3. Permanent Change of Tensile Behaviour after Enduring Large Deformations

The repeatedly discussed switch of tensile behaviour of the polyMEA nanocomposites, from ‘plasto-elastic’ to ‘real elastomeric’, as consequence of intense deformations, is an interesting property, which was investigated in a more detailed way. The results of additional explanatory (mechanical-history-dependent) tensile tests are summarized in Figure 9 (additional results, namely for 0R-15, are shown in Appendix A).

In Figure 9, the second tensile run was always evaluated using the sample’s new geometry, which was acquired due to residual deformation resulting from the previous run.

The results in Figure 9a (example of 4R-15T) illustrate, that a **moderately large deformation**—ca. ½ of the elongation at break in the given case—**does cause only a limited ‘elasticization**’ of the so-treated sample, which was previously in the ‘as prepared’ state. If this sample is stretched until disruption in the second run (as shown in Figure 9a, green curve), and not only to its previous maximum length (as done in the cyclic tests shown further above in Figure 7), its behaviour is at first elastic-like (analogy to Figure 7). But after significantly exceeding the previous maximum absolute length “*L_max*” in “Figure 9a—1st run”, which corresponds to a point somewhat preceding the point “b” in “Figure 9a—2nd run”, the profile of the 2nd-run-curve (in Figure 9a) changes to a ‘plastic-like’ one (plateau with necking effects), which then persists until sample disruption. Some orientation-stiffening can be suspected near the end of this 2nd-run-curve, where the stress somewhat increases with increasing elongation (similarly like in the ‘plasto-elastic’ curve of the ‘as prepared’ nanocomposite in Figure 6c further above). An interesting detail is, that the specific deformation region, where the slope of the 2nd-run-curve somewhat flattens for the first time, labelled with “a” in Figure 9a, is relatively close to the length value (in mm) of the ‘true’ residual deformation endured after the first run (point “L_re” in Figure 9a). An analogous feature also is observed in the cyclic curves of the second hysteresis run for all the polyMEA nanocomposites in the further-above discussed Figure 7f–h (where the maximum elongation in the second cycle was limited, however).

In the experiments shown in Figure 9b, the **complete transition from ‘plasto-elastic’ to ‘real elastomeric’** tensile behaviour was achieved: In the first run, a tensile curve was measured for a 4R-15T specimen in the ‘as prepared state’, until its disruption (black curve, plastic-like). Immediately thereafter, a fragment of the disrupted specimen was again subjected to tensile testing until disruption (green elastomeric-like curve, the steeper one). In another experiment, a fragment of the disrupted sample from the initial (‘plasto-elastic’) tensile test was left to rest for 4 months. After such a rest, this fragment was subjected to a tensile test until disruption, thus yielding the red curve in Figure 9b (elastomeric, but flatter than the green curve). It can be seen, that **after enduring the maximum possible stretching, the polyMEA nanocomposite is converted into an exclusively elastomer-like-behaving material**. Also, no necking during tensile tests is observed anymore after such a previous treatment. In case that the sample went through a longer rest time after the first disruption, the shape of its second tensile curve (Figure 9b: red curve) is practically identical with the ‘immediate second tensile test’ (Figure 9b: green curve), but the slopes at larger deformations are visibly flatter in the ‘relaxed sample’ (red curve). A similar flattening was observed also in Appendix A in case of repeated hysteresis tests with a 30-min-delay for relaxation before the second cycle. For comparison, the ‘geometry-corrected’ maximum elongation in the second deformation cycles in Appendix A corresponds to ca. 100% in Figure 9.

If **toughness** is **considered**, it can be estimated from Figure 9b, that the curve integral of the ‘immediate second disruption test’ (Figure 9b: green curve) yields a ca. 2 times higher toughness, and a similar but somewhat smaller extensibility than the curve (black) of the ‘as prepared’ sample. In case of the ‘relaxed sample’ (red curve), its toughness is practically identical like in case of the ‘as prepared’ sample (see also toughness values in Appendix A).

If the **‘initial moduli’** are compared, which are observed **at the smallest deformations** in Figure 9, they appear to be very similar. Small-strain-torsion experiments (see Appendix A) verified this finding: the shear moduli were **always practically identical** for a given composition, **independently of its previous mechanical history**, in contrast to local moduli (slopes) in regions of large deformations, which are well-visible in Figure 9b.

The partial or full transformation of the polyMEA nanocomposites from ‘plasto-elastic’ to ‘real elastomeric’ state can be explained by the already mentioned reversible hydrogen bridging between polyMEA and silica, and by its temporary destruction and subsequent reconnection and ‘reorganization’ (which is fast but not immediate, see above sample relaxation effects) during large deformations of the ‘as prepared’ (‘plasto-elastic’) specimens. The mechanism will be discussed in detail in the section “Assumed mechanism of the change in the tensile behaviour” (see further below).

#### 3.2.4. Hysteresis Tests of ‘Elasticized’ Samples

The transformation of polyMEA nanocomposites from ‘plasto-elastic’ into ‘real elastomeric’ materials was further studied by hysteresis tests, analogous to the ones presented in Figure 7. Samples which underwent ‘full elasticization’ via stretching until disruption, and which subsequently went through a period of rest (4 months), were used for this purpose. The Appendix A summarizes the results of these investigations for the most important silica-filled nanocomposites 0R-15T and 4R-15T.

The sets of hysteresis curves of elasticized samples in Appendix A (two subsequent cycles were recorded with each sample) are superficially similar to the ones obtained for ‘as prepared’ samples in the same type of test (see further above, Figure 7c,d,g,h). However, the curves of the first loading cycles of the ‘elasticized’ specimens are steeper than in case of the ‘as prepared’ ones: they are elastomer-like instead of plastic-like: no plateau in case of the first run in Appendix A, in contrast to Figure 7c,d further above.

If the second deformation cycle is evaluated using the corrected starting geometry (Appendix A), its elastic-like curves become markedly steeper, with initially identical (4R-15T), or even with slightly steeper (0R-15T) slopes at the lowest strains, than in case of the first run. The latter steepness of the ‘immediate second-run-cycles’ seems to partly stem from a non-equilibrium state, similarly like the steeper curve of the immediately repeated tensile destruction test in Figure 9b.

The **stronger elastomeric character after previous stretching** until destruction is illustrated also by the **improved degrees of deformation recovery**, which can be extracted from the curves in Appendix A:

The highest elastic recovery is observed in the nanocomposite **0R-15T** (polyMEA with 15% of silica) (see Appendix A), which was the most plastic of the polyMEA nanocomposites in the ‘as prepared’ state. Upon unloading from the first stretching, 64% of the initial deformation is recovered, and this value improves to 75% in the moment when the immediate second loading cycle starts; 61% is recovered immediately upon unloading this second cycle. The recovery in the ‘as prepared’ state was 43, 52, and 39% in analogous situations.

‘Elasticized’ **4R-15T** displays a similar but slightly smaller recovery like ‘elasticized’ 0R-15T, namely 60, 71, and 57% in analogous situations like above. (In the as-prepared state, the values were 48, 55, and 43%, respectively).

The recovery value at the start of the second loading cycle seems to be the most representative one for assessing the ‘degree of elasticity’. Hence, 75 and 71% are achieved in 0R-15T and 4R-15T, respectively (vs. 52 and 55% in the ‘as prepared’ state). The stronger improvement in case of 0R-15T also correlates with its steeper cycle in Appendix A.

The hysteresis results further confirm a thorough ‘elasticization’ of the polyMEA nanocomposites, which was documented in the history-dependent tensile tests in Figure 9, as well as a residual small plasticity (creep), which seems to be connected with the physical nature of the crosslinking. The differences between the initial and the geometry-corrected second loading cycles suggest, that small and reversible reorganizations of hydrogen bridging between matrix and filler (which are discussed just below) play a role also in the permanently elasticized state (the filler-free neat polyMEA practically does not display such differences, see Figure 7a,e).

#### 3.2.5. Assumed Mechanism of the Change in the Tensile Behaviour

##### Molecular Level

The mechanism of the further-above discussed huge energy-absorption ability in the ‘as prepared’ state of the polyMEA nanocomposites, as well as the phenomenologically related ‘elasticization’ seems to be connected with the phase separation processes during their synthesis. Analyses of morphology of ‘elasticized’ samples, which are discussed just below (and illustrated by Appendix A), did not indicate any marked changes in the nanofiller patterns, but only slight ones. It seems that the nanofillers, especially the strongly hydrogen-bridging silica, are not optimally covered by the polymer molecules in the moment of the phase-separation: Concerned are especially the secondary adsorptions (H-bridging) of the free-to-move polyMEA segments (Appendix A). This situation is postulated to be a consequence of the relatively abrupt phase separation during the polymerization (which was discussed in context of the further-above-shown Figure 2). Also the formation of a part of polymer-polymer entanglements (see Appendix A) might be adversely affected by the abrupt precipitation of polyMEA from the aqueous solution. During the extensive deformations in the course of tensile tests (and during a short rest time after them), the supramolecular elastic network structure (see further-above Figure 4g–i and Appendix A) seems to undergo repeated dissociation and reconnection processes, which lead to subsequently more efficient physical crosslinking (‘switching to real elastomeric behaviour’), but only to slight alterations in morphology. The irreversible break-up of the initially present ‘non-ideal’ but mechanically strong morphological features then corresponds to the large hysteresis areas of the initial deformation cycles of the nanocomposites in the ‘as prepared’ state (see Figure 7 further above). The originally present partly plastic internal structures do not seem to re-appear (due to the higher stability of the newly connected ‘optimal’ polymer-filler crosslink system): The tensile curve measured immediately after previous disruption, as well as the one measured after disruption and 4 months of rest, both have the same shape (after rest, the slopes at large deformations are even flatter and not steeper).

The elasticity of the material ‘homogenized’ by large deformation, as well as the stiff-plasto-elastic-like behaviour of the ‘as prepared’ one, might both be of value for practical applications. The strong affinity of in-situ-nano-silica to polyMEA chains also might be responsible for the absence of self-healing in the studied silica-containing polyMEA nanocomposites (disentangled chains rapidly adsorb on SiO_2_), in contrast to at least some polyMEA/clay systems (see [66]).

##### Morphology Changes Visible by TEM

The morphology of the samples, which were subjected to stretching until destruction, and hence to ‘internal homogenization’, but probably also to some degree of aligning, was investigated by TEM (see Appendix A). The stretching direction was always in the plane of the recorded images. It can be seen from the comparison in Appendix A, that at the first glance the morphology did not markedly change. However, at closer inspection, it can be noted (4R-15T in Appendix A vs. Appendix A) that most of the sharp lines disappear, which were associated with groups of clay platelets, so that the latter seem to be less often oriented perpendicularly to the image plane (hence more often parallel). Also, the largest groups of filler are broken to more regularly sized smaller ones (this correlates with reduced plasticity). But in case of the sample with 15% of SiO_2_ as the only filler, differences are practically invisible. In case of the sample filled exclusively with clay (4% RDS), the overall pattern also does not change, but the platelets seem to be more often oriented parallel to the image plane, in which also laid the stretching axis.

##### Morphology Changes Visible by X-ray

If the effect of stretching until break on the morphology is observed by X-ray diffraction (see Appendix A), the changes in pattern also are moderate, but more clearly visible than by TEM. In case of samples containing 15 wt.% of silica (with or without clay), a change in the position of the maxima caused by the SiO_2_ phase can clearly be observed: the maximum generated by the smallest secondary domains moves to higher angles (smaller domain size) in a nearly identical way in both the mentioned samples. In case of the sample filled by clay only (4 wt.%), no significant pattern change can be observed in the 1D diffractograms in Appendix A. But if the 2D images used to obtain the scattering patterns are evaluated (Appendix A), some orientation effects can be recognized in case of the clay-containing samples, especially a variation in the intensity of diffraction rings (marked in 4R-0T: see Appendix A). This confirms the mentioned impression gained from TEM (see Appendix A vs. Appendix A). On the other hand, the exclusively SiO_2_-filled samples do not display notable anisotropy in 2D diffractograms after they endured stretching.

Morphology analyses hence confirm, that changes in filler patterns are small after the ‘elasticization’ of the nanocomposites, which resulted from very large deformations. This also means, that the reorganization of hydrogen bonding between SiO_2_ and polyMEA (and of the less energetic polyMEA-clay interactions) is the main mechanism of the ‘elasticization’ of the polyMEA nanocomposites, as well as of the energy absorption by them in the ‘as prepared’ state.

### 3.3. Thermo-Mechanical and Thermal Properties

#### 3.3.1. Glass Transition Temperatures and Moduli as Observed by DMTA

The thermo-mechanical properties, especially the temperature-dependent moduli, as well as the glass transition temperatures (*T*_g_) of the prepared polyMEA nanocomposites, and also of reference materials, were investigated by means of **dynamic-mechanical thermal analysis (DMTA)**. The results are summarized in Figure 10, as well as in Table 3.

It can be observed in Figure 10, that the moduli in the rubbery region are dramatically increased by the nanofillers, by up to three orders. At the same time, the glass transition temperatures (see Table 3) are only slightly altered and remain very close to the value observed for neat polyMEA (−25.6 °C).

In case of the **nanocomposites filled exclusively with silica** (see Figure 10b and Table 3), the maximum of the *tan*(*delta*) *= f*(*T*) curve, which was used to define the *T*_g_ value, slightly shifts by a few degrees to higher or lower temperatures in a complicated trend, as the content of silica is raised. This can be assigned to competing effects, such as polyMEA immobilization by large filler-rich domains (see morphologies in Figure 4 further above), vs. the effect of increased local irregularities in the arrangement of neighbouring polymer chains, as consequence of hydrogen bridging between polyMEA and the primary silica particles. As the SiO_2_ content is raised, the *tan*(*delta*) peaks become lower, and a shoulder at their high-temperature slope becomes increasingly prominent. At 30 wt.% of the filler, the ‘peak’ is very broad, it is rather a step in the *tan*(*delta*) *= f*(*T*) curve, which does not display a well-defined maximum, but a high plateau in the range from ca. −28 to ca. −1 °C. The broadening of the peak towards higher temperatures is a consequence of an increasingly prominent fraction of partly immobilized polyMEA in the neighbourhood of the filler, as nano-SiO_2_ increasingly percolates the whole sample (see TEM in Figure 4c,f). The decrease of peak height can be correlated with increasingly high values of *G’* (see Figure 10a) relatively to *G*″ in the transition region. Another interesting detail is, that *tan*(*delta*) values in the rubbery region decrease (in comparison to polyMEA, see Figure 10b) as the filler amount increases. This might be correlated with increased crosslinking between SiO_2_ and polyMEA, as well as with the increased fraction of the rigid filler (both → higher elasticity (*G*′)). If going from 15 to 30% of nano-SiO_2_, the *tan*(*delta*) values in the rubbery region slightly increase (relatively to 10%, but not to 5% of SiO_2_), thus indicating over-filling and additional friction caused by it.

The **doubly filled polyMEA nanocomposites** display similar trends concerning the slight shifts in their *T*_g_ values, like the ones filled with silica only. Addition of 4 wt.% of RDS clay to polyMEA causes a broadening of the *T*_g_ peak, which displays a high plateau region with two distinguishable summits: at −26.7 and at −17.6 °C. They correspond to nearly unaffected polyMEA, and to polyMEA immobilized by the proximity of the relatively extended 2D patterns of clay nanoplatelets (see TEM, Figure 4d further above), respectively. The value of *tan*(*delta*) in the rubbery region is slightly increased by the clay (effect of complex and entangled supramolecular structure). Addition of nano-silica leads to simple peak shapes with a single maximum and also with a high-temperature shoulder, which at 15 wt.% of silica (in addition to 4 % of clay) evolves into a second broad and flat maximum centred around +3 to +5°C (polyMEA immobilized by silica). The value of *tan*(*delta*) in the rubbery region decreases with increased silica content in the doubly filled elastomers (tested from 5 to 15% of SiO_2_ in combination with 4% of clay). Hydrogen bridging by the small nano-SiO_2_ particles seems to reduce inter-molecular friction, and to favour a more uniform supramolecular structure.

In Figure 10a,c, the **tremendous effect of the content of the incorporated nanofillers on the moduli** of the polyMEA-based elastomers in the rubbery state can be analysed. It can be noted, that the most dramatic changes are achieved by the addition of small amounts of silica or of clay to neat polyMEA: If 5% of SiO_2_ or 4% of RDS are added, the modulus rises by ca. one order, namely 7.9 or 19.3 times, respectively (see Figure 10 and Table 3). It could be suspected, that in the low-filled nanocomposites, a fraction of the smallest inorganic particles is not grouped into the observed domains, and is thus free for maximally efficient interactions with polyMEA (due to a high specific surface area). A similar modulus increase, by ca. one more order is achieved if small amounts of both fillers are combined (4% of RDS + 5% of SiO_2_, synergy effect), or if the content of SiO_2_ is increased from 5 to 15%.

The synergy of the fillers is less pronounced at high filler loadings, e.g., if 15% of SiO_2_ are complemented by 4% of clay: only a moderate increase of modulus is observed (see Figure 10 and Table 3), similarly like in case of increasing the content of SiO_2_ (as exclusive filler) from 10 to 15%. Hence, at 15%, the hydrogen bridging between polyMEA and the fine-grained nano-silica seems to reach saturation. However, a dramatic increase of modulus again is observed, if the silica content is increased from 15 to 30%, which raises the modulus by one order, to a value close to 10^8^ Pa. Such a high modulus, as well as the fragility of the sample suggests an effect of percolating rigid filler (which also is supported by the *tan*(*delta*) *= f*(*T*) curve in Figure 10b).

#### 3.3.2. Thermal Transitions as Observed by DSC

The glass transitions, and especially the change in heat capacity associated with them was investigated by means of differential scanning calorimetry (DSC). Interesting results were obtained and are summarized in Table 4. The results are also illustrated by the DSC trace of the sample 0R-15T in Appendix A; the very similar traces of the other samples are shown in Appendix A.

It can be observed in Appendix A, that the glass transitions of all the studied polyMEA-based elastomers are very well visible in DSC traces, as distinct and relatively sharp steps in heat capacity. This distinct DSC behaviour seems to be an intrinsic property of polyMEA. The *T*_g_ values observed by DSC expectedly displayed little dependence on the amount of the incorporated filler, similarly like in case of DMTA. In case of DSC, the sensitivity to the filler amount was even weaker: neat polyMEA displayed *T*_g_ = −30.8 °C in DSC, while the *T*_g_ values for the most differing samples were lower or higher by maximally 1 °C. Also the shapes of the DSC curves were nearly identical. Similarly like in case of DMTA, the samples with the very highest content of filler(s) displayed the highest values of *T*_g_.

In contrast to the temperature of glass transition, the height of the associated step in heat capacity displayed a strong dependence on the amount of the inorganic nanofillers: if going from neat polyMEA to the sample 4R-15T (19 wt.% of combined fillers), the height of the step decreased by ca. 50%. This change occurred approximately linearly with increasing amount of combined inorganic fillers: they namely do not contribute thermally to the glass transition itself.

### 3.4. Stability against Oxidative and Thermal Degradation (TGA)

The stabilization of the polyMEA elastomers against oxidation and thermolysis, which is provided by the inorganic nanofillers, was evaluated by means of thermogravimetric analysis (TGA). The most important results are summarized in Table 5 and Table 6, and shown in Appendix A.

Generally, it was found, that the nanofillers exerted only a small influence on the thermal stability in air and in nitrogen, at least in the investigated range of filler loadings. This small nanofiller effect was stronger in case of the oxidative degradation in air.

In case of the **nanocomposite elastomers filled with silica only**, it can be seen, that the small amount of 5 wt.% of nano-SiO_2_ slightly stabilizes the elastomer against oxidation by air (most probably via less permeable (stiffer) H-bridged supramolecular structure): both the decomposition onset, as well as the temperature of the maximum oxidation rate are very modestly improved. At 15% of SiO_2_, however, the nanocomposite is slightly destabilized, especially concerning the temperature of the maximum oxidation rate. Possibly, the nano-SiO_2_ filler itself supports the diffusion of oxygen into the sample, because no analogous destabilization is observed under nitrogen.

In case of the **clay- and doubly filled nanocomposites**, the clay filler—if it is alone (e.g., in the sample 4R-0T)—displays the strongest stabilizing effect against oxidation by air among the studied filler combinations. Increasing the content of nano-SiO_2_ added with the clay causes a decrease in the mentioned stabilizing effect. In case of 4 wt.% of clay and 15 wt.% of silica (4R-15T), the nanocomposite is already less stable than the neat matrix. The anti-oxidative action of the RDS clay likely is based on the barrier effect of the inorganic nano-platelets.

In case of the **TGA analyses carried out under nitrogen** (see Table 6, as well as Appendix A), the differences between the nanocomposites are very small. Nevertheless, in case of nano-SiO_2_ as the exclusive filler, a moderate stabilizing effect can be observed, which increases with filler content. This is probably due to matrix immobilization by the SiO_2_ filler via H-bridging (the above-mentioned support of oxygen transport by SiO_2_ does not play a role in the inert atmosphere). In case of the clay-filled elastomers, the clay seems to moderately catalyse the thermolysis (while the above-mentioned barrier effect (anti-oxidative stabilization) is useless under inert atmosphere)—the temperature of decomposition onset is lowered by it—but the overall stability is similar like in case of neat polyMEA. Addition of 5 wt.% of nano-SiO_2_ (which tends to concentrate around the clay platelets) cancels the decomposition catalysis by clay, and up-shifts the onset temperature, as well as the temperature of the maximum oxidation rate in comparison to neat polyMEA. With 15 wt.% of SiO_2_ in addition to 4 wt.% of clay, the stabilization effect of SiO_2_ somewhat recedes, possibly due to the already mentioned (tensile properties, DMTA) ‘over-filling’ effect.

## 4. Conclusions

This work was dedicated to the synthesis and exploration of novel supramolecularly assembled polyMEA nanocomposite elastomers containing a new filler, namely small in-situ-formed nano-silica particles (3–5 nm), which served as macro-crosslinker of ultra-high-molecular-weight chains, which additionally tend to physical crosslinking via entanglements; doubly filled polyMEA nanocomposites containing clay nano-platelets and nano-silica were also prepared, and the synergy of both fillers was evaluated, which led to some interesting effects.

The in-situ-formed silica nanospheres proved to be a very efficient physical crosslinker in the self-assembled elastomeric architecture (via hydrogen bridging): they lend excellent material properties to the studied nanocomposites, especially very high moduli, yield stresses, toughness and tensile strength, in combination with ultra-extensibility; both soft and permanent crosslinks play a role; all the polyMEA nanocomposites displayed an interesting ‘switching’ in tensile behaviour: in the post-synthesis state, they exhibit ‘plasto-elasticity’ (only 50% retraction of deformation, and necking during deformation), but after the first extensive stretching, they become ‘real elastomers’; this was assigned to polyMEA-nanofiller interactions, especially H-bridging with nano-SiO_2_, and to their fine-scale rearrangement during the stretching; the glass transition temperature, as well as the thermal stability (TGA) is only slightly influenced by the nanofillers; but the glass transition is very distinct in both mechanical- and DSC tests.

The studied polyMEA/silica nanocomposites offer interesting application possibilities, due to the combination of their excellent mechanical properties with the known bio-compatibility of the components: in the biomedical field, they could be of interest as implant material, artificial tissues, or cell scaffolds; in the technical field, they could play bio-analogous roles in robotics; the high transparency is an additional valuable property of the studied materials.

## Figures and Tables

**Figure 1 polymers-13-04254-f001:**
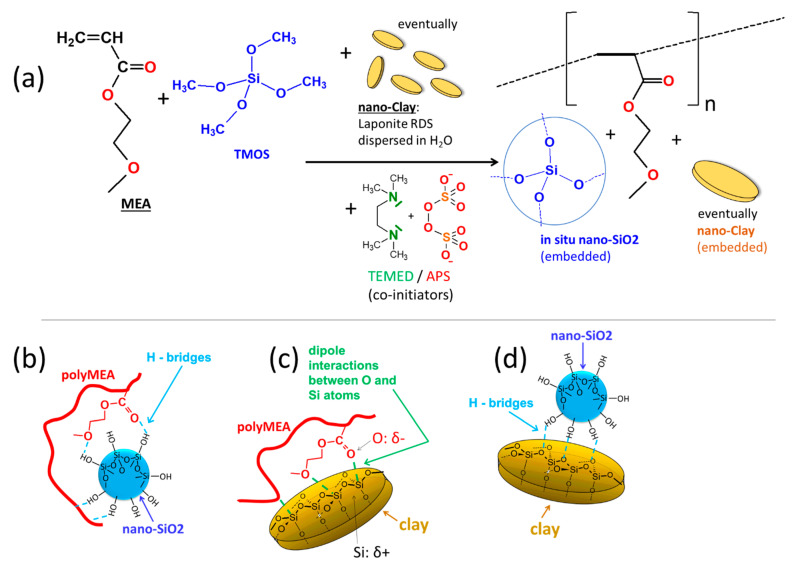
(**a**) Synthesis of the studied nanocomposites; (**b**–**d**) Interactions between the phases involved in the studied nanocomposites: (**b**) polyMEA interactions with nano-SiO_2_ via hydrogen bridging; (**c**) interactions between nano-SiO_2_ and clay nano-platelets also via hydrogen bridging; (**d**) interactions between clay nano-platelets and polyMEA via dipole-dipole interactions.

**Figure 2 polymers-13-04254-f002:**
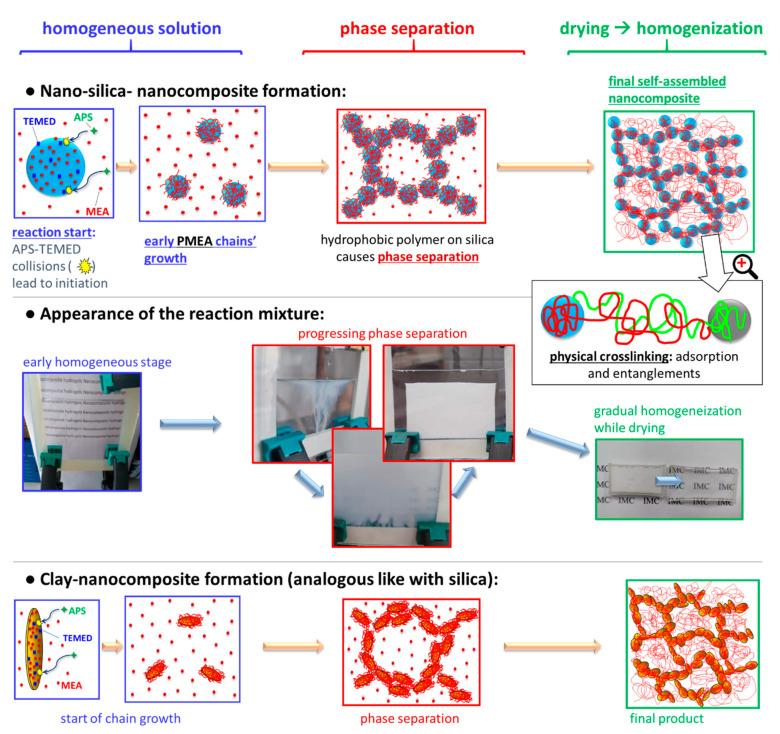
Development of the phase structure of the studied nanocomposites during their synthesis.

**Figure 3 polymers-13-04254-f003:**
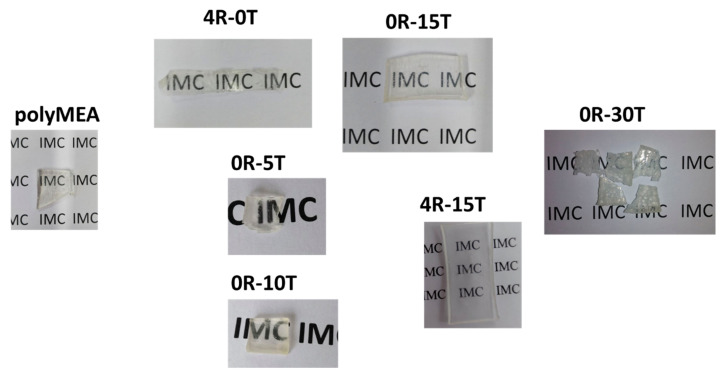
Outward appearance of the most important studied products.

**Figure 4 polymers-13-04254-f004:**
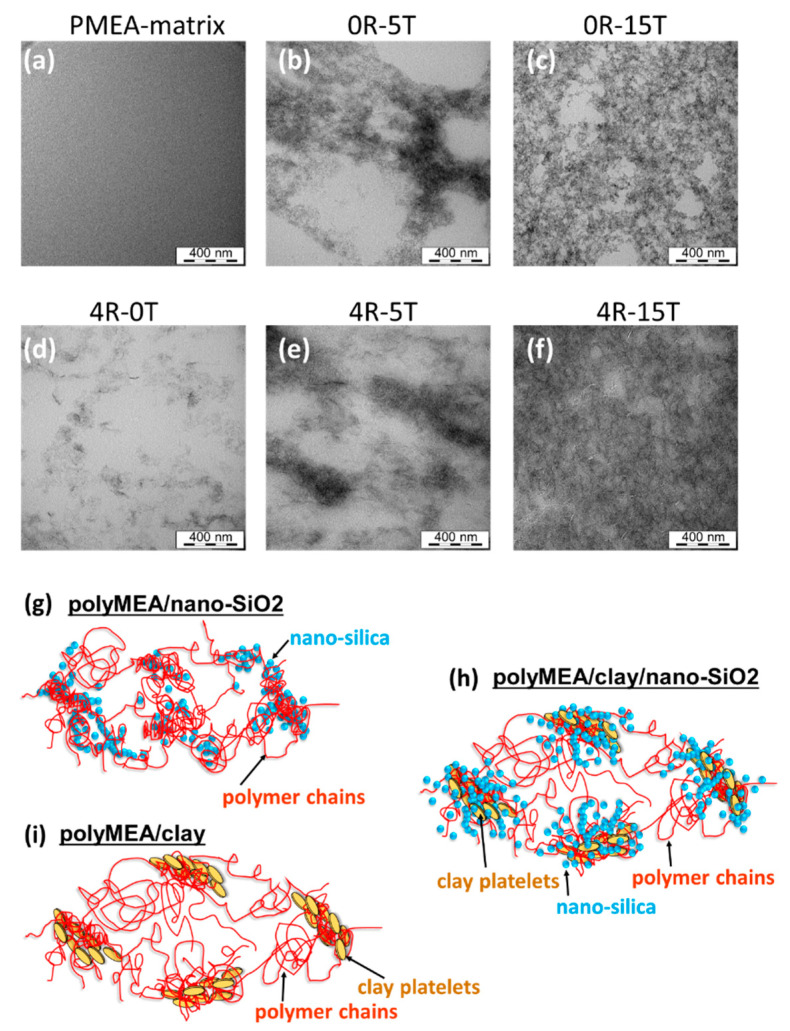
(**a**–**f**) Morphology of the most important compositions of the studied nanocomposites; (**g**–**i**) postulated supramolecular structures of the studied nanocomposite networks: (**g**) polyMEA/nano-SiO_2_; (**h**) polyMEA/clay/nano-SiO_2_; (**i**) polyMEA/clay (reference material); high-resolution TEM micrographs are shown in Appendix A.

**Figure 5 polymers-13-04254-f005:**
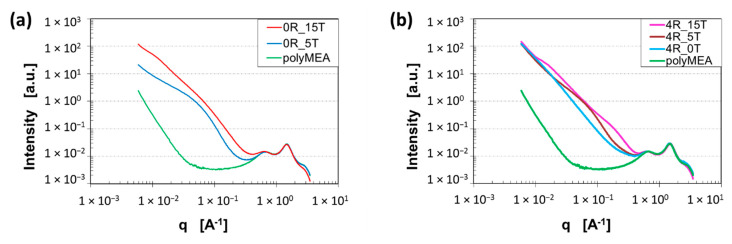
X-ray scattering patterns: (**a**) effect of nano-SiO_2_; (**b**) effect of combined fillers nano-SiO_2_ + clay.

**Figure 6 polymers-13-04254-f006:**
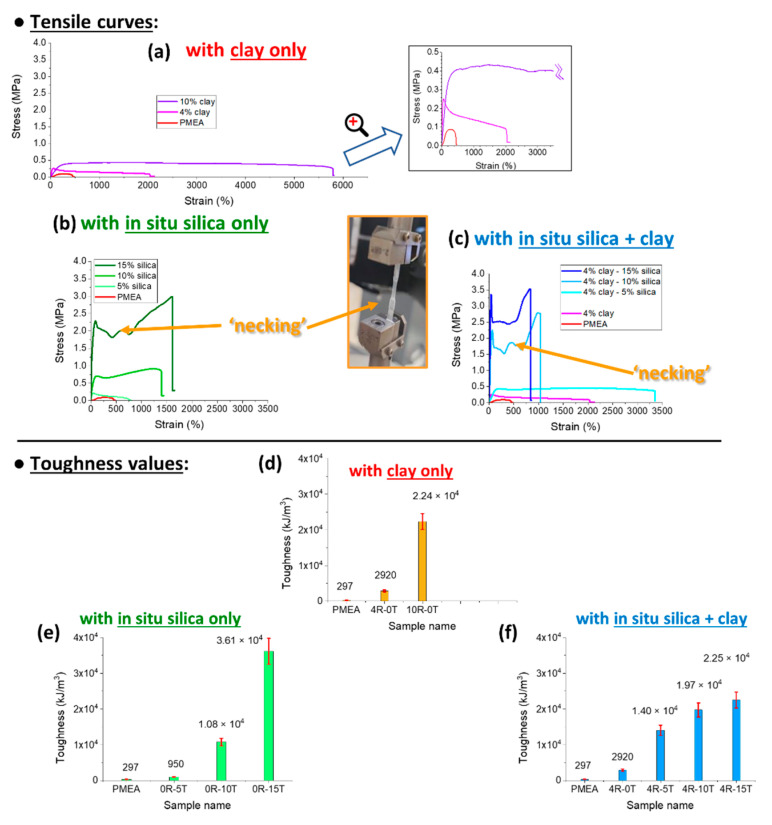
(**a**–**c**) Tensile curves of (**a**) reference materials polyMEA (neat), polyMEA/4% clay, and polyMEA/10% clay; (**b**) polyMEA/nano-SiO_2_ systems containing 5, 10 and 15% of filler; (**c**) polyMEA/clay/nano-SiO_2_ systems containing always 4% of clay + additionally 0, 5, 10 and 15% of nano-SiO_2_; (**d**–**f**) Tensile toughness values of the same groups of samples, the curves of which are shown in (**a**–**c**).

**Figure 7 polymers-13-04254-f007:**
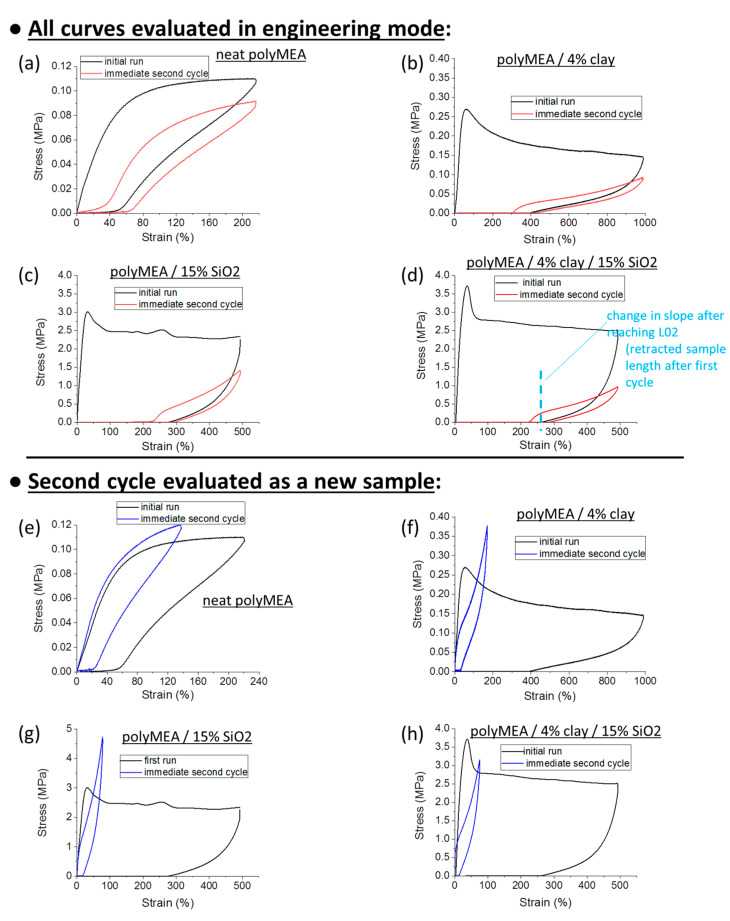
(**a**–**d**) Hysteresis curves up to large deformations of (**a**) neat polyMEA; (**b**) polyMEA/4% clay; (**c**) polyMEA/15% SiO_2_; and (**d**) polyMEA/4% clay/15% SiO_2_; (**e**–**h**) same experiments, but with evaluation of the second cycles as new specimens with new initial geometry (‘true material properties’ in the second cycle).

**Figure 8 polymers-13-04254-f008:**
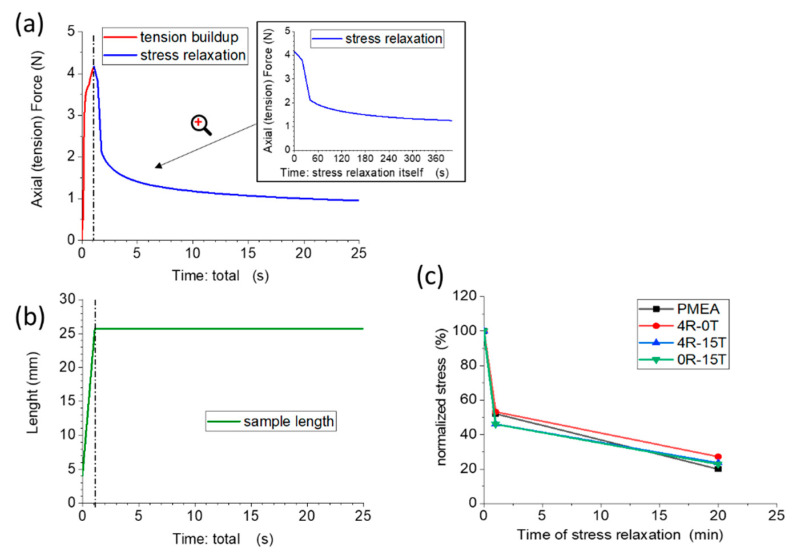
Stress relaxation experiment with an exemplary polyMEA-based nanocomposite elastomer (4R-15T): (**a**) graph of stress build-up (as consequence of mechanical stretching, red line) and of subsequent relaxation (during the rest period, blue line); (**b**) length of the specimen between the clamps of the analysing machine during this experiment; (**c**) comparison of the course of the relaxation of normalized stress (100% = initial value) for the most important of the studied elastomers and for the related reference materials.

**Figure 9 polymers-13-04254-f009:**
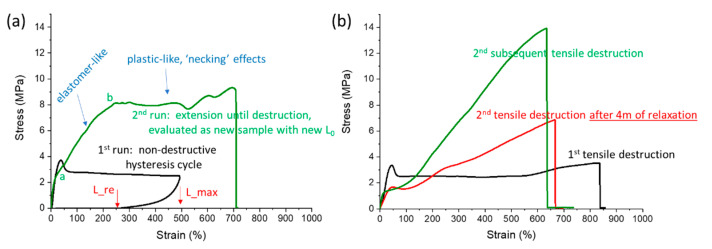
Effect of mechanical deformation history on tensile characteristics, on the example of the doubly filled nanocomposite 4R-15T: (**a**) non-destructive hysteresis loop up to ca. 50% of the elongation at break—this leads to elastic behaviour in subsequent cycles with the same maximum elongation in mm (see Figure 7)—is followed immediately by a more extensive second tensile test which runs until sample disruption; (**b**) successive tensile tests until disruption, first with an ‘as prepared’ sample, thereafter with one of the ‘stretching-treated’ pieces obtained after disruption—either immediately after the first test, or after 4 months of rest.

**Figure 10 polymers-13-04254-f010:**
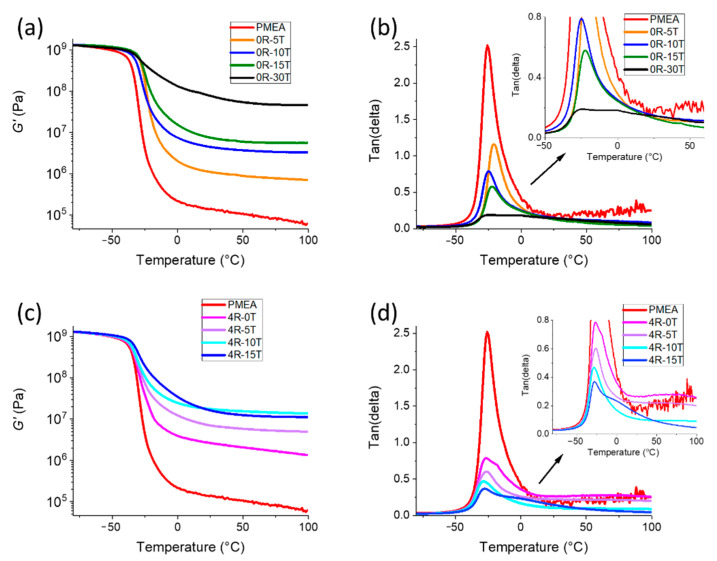
Dynamic-mechanical thermal analysis (DMTA) of (**a**,**b**) polyMEA and its nano-silica-filled derivatives and (**b**,**c**) of nanocomposites of polyMEA doubly filled with clay and nano-silica: (**a**,**c**) curves of the temperature-dependence of storage shear moduli; (**b**,**d**) curves of the temperature-dependence of the loss factor (tan(delta)).

**Table 1 polymers-13-04254-t001:** Amounts of components for the syntheses of the studied nanocomposite elastomers. (abbreviations: MEA = 2-Methoxyethyl acrylate monomer, polyMEA = neat (filler-free) polymerized MEA, TMOS = tetramethoxysilane, TEMED = N,N,N′,N′-tetramethylethylenediamine, APS = ammonium persulfate, RDS = product name of the clay used by the supplier).

Sample Name	Water	Clay RDS	MEA	*n* MEA	TMOS	*n* TMOS	TEMED	*n* TEMED	1% APS_aq_	*n* APS
	g	g	g	mmol	g	mmol	g	mmol	g	mmol
polyMEA	42.312	0.000	5	38.420	0.000	0.000	0.0625	0.538	3.814	0.167
0R-5T	41.652	0.000	5	38.420	0.661	4.342	0.0625	0.538	3.814	0.167
0R-10T	40.909	0.000	5	38.420	1.404	9.220	0.0625	0.538	3.814	0.167
0R-15T	40.084	0.000	5	38.420	2.228	14.638	0.0625	0.538	3.814	0.167
0R-30T	36.891	0.000	5	38.420	5.421	35.616	0.0625	0.538	3.814	0.167
4R-0T	42.289	0.230	5	38.420	0.000	0.000	0.0625	0.538	3.814	0.167
4R-5T	41.629	0.230	5	38.420	0.661	4.342	0.0625	0.538	3.814	0.167
4R-10T	40.886	0.230	5	38.420	1.404	9.220	0.0625	0.538	3.814	0.167
4R-15T	40.061	0.230	5	38.420	2.228	14.638	0.0625	0.538	3.814	0.167

**Table 2 polymers-13-04254-t002:** Content of all inorganic fillers determined via ash analysis (abbreviated sample names are all listed in Table 1; “R” symbolizes the clay content, “T” the SiO_2_ content: e.g., “4R-5T” means a nanocomposite with 4 wt.% of clay (RDS) and 5 wt.% of SiO_2_).

Sample	wt.% of Ashes
0R-5T	4.92
0R-15T	14.75
4R-0T	3.79
4R-5T	7.43
4R-15T	17.0

**Table 3 polymers-13-04254-t003:** Shear moduli G’ (at room temperature = 25 °C) and glass transition temperatures (T_g_, determined as x-positions of the maxima of the tan(delta) = f(T) curves) of the samples whose DMTA profiles are shown in Figure 10 (abbreviated sample names are all listed in Table 1; “R” symbolizes the clay content, “T” the SiO_2_ content: e.g., “4R-5T” means a nanocomposite with 4 wt.% of clay (RDS) and 5 wt.% of SiO_2_; “polyMEA” = poly(2-Methoxyethyl acrylate)).

Sample	*G*′ [MPa]	*T*_g_ [°C]
neat polyMEA	0.137	−25.6
0R-5T	1.08	−20.8
0R-10T	4.36	−24.9
0R-15T	7.51	−22.4
0R-30T	74.5	broad plateau peak: from −28 to ca. −1
4R-0T	2.64	peak with flat summit −26.71 and −17.63 (smaller)
4R-5T	6.97	−26.7
4R-10T	17.7	−28.8
4R-15T	16.4	−28.4 & second very broad peak centered near +3 to +5

**Table 4 polymers-13-04254-t004:** Temperatures of glass transition (T_g_, determined as T of mid-step in heat capacity), as well as the associated changes in heat capacities for the most important among the studied nanocomposite elastomers, and for some reference materials (abbreviated sample names are all listed in Table 1; “R” symbolizes the clay content, “T” the SiO_2_ content: e.g., “4R-5T” means a nanocomposite with 4 wt.% of clay (RDS) and 5 wt.% of SiO_2_; “polyMEA” = poly(2-Methoxyethyl acrylate)).

Sample	*T*_g_ [°C]	Change in Heat Capacity [J g^−1^ K^−1^]
neat polyMEA	−30.8	0.604
0R-5T	−31.4	0.584
0R-15T	−30.3	0.429
4R-0T	−31.2	0.617
10R-0T	−31.3	0.465
4R-5T	−31.7	0.556
4R-15T	−29.7	0.324

**Table 5 polymers-13-04254-t005:** TGA analysis of the most important samples in air: Temperatures of decomposition maxima as determined by differential thermogravimetry, as well as trends of decomposition (abbreviated sample names are all listed in Table 1; “R” symbolizes the clay content, “T” the SiO_2_ content: e.g., “4R-5T” means a nanocomposite with 4 wt.% of clay (RDS) and 5 wt.% of SiO_2_; “polyMEA” = poly(2-Methoxyethyl acrylate)).

Sample	Temperatures of Decomposition Maxima [°C]	Trends of Decomposition
neat polyMEA	411.7	(reference)
0R-5T	(shoulder 396.3); 427.4	later onset, stabilized at all *T*
0R-15T	369.3; 425.11 (smaller)	destabilized: later onset but thereafter more readily oxidized
4R-0T	429.8	stabilized, most strongly among the tested samples
4R-5T	402.8	maximum decomposition rate at lower temperature, but more stable than matrix until 392 °C
4R-15T	361.8 and 427.1	destabilized, also earlier onset

**Table 6 polymers-13-04254-t006:** TGA analysis of the most important samples in N_2_: Temperatures of decomposition maxima as determined by differential thermogravimetry, as well as trends of decomposition (abbreviated sample names are all listed in Table 1; “R” symbolizes the clay content, “T” the SiO_2_ content: e.g., “4R-5T” means a nanocomposite with 4 wt.% of clay (RDS) and 5 wt.% of SiO_2_; “polyMEA” = poly(2-Methoxyethyl acrylate)).

Sample	Temperatures of Decomposition Maxima [°C]	Onset of Decomposition
neat polyMEA	417.5	(reference)
0R-5T	425.5	later onset, more stable
0R-15T	430.3	later onset, more stable
4R-0T	425.8	earlier onset, but comparable stability
4R-5T	429.8	visibly later onset, moderately stabilized
4R-15T	427.1	ca. same onset like matrix, comparable stability

## Data Availability

The data presented in this study are available on request from the corresponding author.

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
