# Peer review of "Novel Tough and Transparent Ultra-Extensible Nanocomposite Elastomers Based on Poly(2-methoxyethylacrylate) and Their Switching between Plasto-Elasticity and Viscoelasticity"

_polymers, 2021, doi:10.3390/polym13234254_

Round 1

Reviewer 1 Report

This paper presented an investigation of a type of poly(2-methoxyethyl acrylate) based elastic nanocomposites. The authors first reviewed the state of the art of elastic polymers and nanocomposites. Then, the materials, fabrication process, and characterization steps were introduced. The authors reported adequate results, but the discussion section can be improved by providing additional clarifications. Please see the detailed comments below. 

  1. There are grammar mistakes and typos in the current draft. Please proofread again before submitting the revised paper.
  2.  The three schematics should be converted to figures and included in the supporting materials. 
  3. In figure 1, each sub figure should have a title or a short description. 
  4. There are 16 figures in the current draft. This is much more than a regular researech article should have. Could the authors combine some of the figures, and reduce the total figures to 8-10? Could some of the figures be moved to the supporting materials? The authors only need to report the most important results in the article content. 

Author Response

Response to comments of Reviewer #1

The authors would like to thank the Reviewer #1 for his time, his attention, thoughtful comments, and valuable suggestions, which helped to considerably increase the reader-friendliness, as well as the attractivity of the submitted manuscript.

Answers:

Reviewer #1:

This paper presented an investigation of a type of poly(2-methoxyethyl acrylate) based elastic nanocomposites. The authors first reviewed the state of the art of elastic polymers and nanocomposites. Then, the materials, fabrication process, and characterization steps were introduced. The authors reported adequate results, but the discussion section can be improved by providing additional clarifications. Please see the detailed comments below.

Answer: The authors are very grateful for the positive assessment and for the useful subsequent suggestions, which helped to streamline the Manuscript and make it more reader-friendly.

Important corrections done in response to the comments of Reviewer #1 are highlighted with turquoise colour in the copy of the revised Manuscript with highlighted changes.

Reviewer #1:

  1. There are grammar mistakes and typos in the current draft. Please proofread again before submitting the revised paper.

Answer: The authors thoroughly checked the revised Manuscript for typos, edit errors, etc., as well as for stylistics and reader-friendliness (especially in case of the more complicated discussions).

Reviewer #1:

  1. The three schematics should be converted to figures and included in the supporting materials.

Answer: All  Schemes were transferred to the Figure category, as suggested. They were partly moved to the Supplementary Information File (former Scheme 1, now SI-Fig. 1), partly joined to a new Figure (former Scheme 2 and 3, as the new Figure 1), and partly embedded in existing illustrations (former Scheme 4, now part of revised Figure 4). The schematic illustrations which were kept as part of the Figures in the main Manuscript file were very important for the Discussion, in which they are often referred-to. The total number of new Figures in the revised Manuscript was reduced as suggested in Point 4. further below.

Reviewer #1:

  1. In figure 1, each sub figure should have a title or a short description.

Answer: The former Figure 1 (now revised Figure 2) was provided with the suggested titles, and it was also stylistically improved for achieving a higher reader friendliness and clarity.

Reviewer #1:

  1. There are 16 figures in the current draft. This is much more than a regular researech article should have. Could the authors combine some of the figures, and reduce the total figures to 8-10? Could some of the figures be moved to the supporting materials? The authors only need to report the most important results in the article content.

Answer: The authors reduced the total number of Figures (including former “Schemes”) to 10, as suggested. Also, the Introduction and Discussion were streamlined and provided by more Headings (for faster orientation in the discussion), some less important comments were removed. This shortening was balanced, however, by explanative texts added in response to suggestions of Reviewer #2. Introduction and Discussion were also thoroughly checked for clarity and reader-friendliness.

Reorganization of Figures:

In the originally submitted Manuscript, there were originally 17 Figures + 4 Schemes = 21 images in total, their number was reduced down to 10 Figures (10 images in total) in the revised Manuscript.

Changes (according to position in original Manuscript):

Scheme 1 --> moved to the Supplementary Information File (SI), as  SI-Fig. 1

Scheme 2+Scheme 3   were joined together, as the new Figure 1

Figure 1 (structure development, phase separation) was kept as new Figure 2, but stylistically improved (titles and separation lines were added, as well as an inlay sub-image about physical crosslinking)

Figure 2 was kept unchanged, as new Figure 3

Figure 3 (TEM, now new Figure 4) was streamlined and merged with Scheme 4 (which explained the results in the Figure), large scalebars were provided for the TEM micrograms, micrograms with larger-scale-views were removed to SI, where also high-resolution versions of all micrograms are provided.

Figure 4 (X-ray, now new Figure 5) was kept but reduced in size: the sub-image Figure 4b was deleted

Figure 5 (X-ray: SAXS region) --> moved to SI,   as new SI-Fig. 4

Figure 6 and Figure 7 (tensile curves, toughness values) were joined:   now new Figure 6

Figure 8 and Figure 10 (cyclic loading tests) were joined:    now new Figure 7

Figure 9 (tensile testing: multiple cycling, effect of rest before repeated cycling) --> moved to SI, as new SI-Fig. 7

(Figure 10: joined with above former Figure 8)

Figure 11 (relaxation) was kept as    new  Figure 8

Figure 12 (effect of deformation history on tensile properties) was kept as    new  Figure 9

Figure 13 (hysteresis tests with samples elasticized by previous stretching) --> moved to SI,   as new SI-Fig. 14

Figure 14 (DMTA)  was kept as    new  Figure 10 (last in Manuscript)

Figure 15  (DMTA: filler effect analysis)   -->  was deleted

Figure 16 (DSC) --> moved to SI,   as new SI-Fig. 18

Figure 17 (TGA overview) --> was deleted (the more detailed  SI-Fig. 19 and 20 are cited instead).

Reviewer 2 Report

The manuscript entitled “Novel tough and transparent ultra-extensible nanocomposite

 elastomers based on poly(2-methoxyethylacrylate), and their 3 switching between plasto-elasticity and viscoelasticity” is a document of interesting subject matter particularly studies done on morphology by the SAXS and TEM are very attractive.

However, it needs some major changes before being accepted. Make the following corrections:

  1. The interpretation of the experimental results should be significantly improved. In other words, pay attention on the interpretation of the experimental results.
  2. The authors discussed the potential of elastomers as efficient polymers for implant technologies, robotics or soft robotics. In this context, other potentials of these nanocomposites, specifically as nano-cement in artificial bones, and scaffolding, should be discussed as well.
  3. It is recommended that the table does not have minor horizontal lines. Please check the tables.
  4. Tables must be independent of legible text. Please check tables, especially Table 1, and include the abbreviations (MEA, TMOS, ...) in the caption below of the table.
  5. In Figure 7, is the results checked statistically? There is a significant difference between analyses? If there is a significant difference statistically, it is recommended to mention it in the caption of the figure.
  6. I am not convince with the introduction section write up. Kindly rewrite this section again.
  7. The nanomaterials can be used as efficient materials in cosmetics. In this context, the authors should discuss the development of novel polyMEA nanocomposite elastomers with mentioned new fillers as dermal fillers (lip and facial fillers, or wrinkle fillers). The following papers must be cited and discussed in the introduction: (DOI: 10.1007/s00289-020-03354-6, DOI: 10.3390/polym13071161,)
  8. On page 4 section 2.1. Materials, provide the purity, grade and made of chemicals used in the present work.
  9. Kindly include few refrence under introduction section. org/10.3390/polym13183153; doi.org/10.3390/nano11051086
  10. On page 11, Figure 3. Morphology of the most important compositions of the studied nanocomposites. Kindly provide the clearly scale bar in each of the TEM images.
  11. Scheme 4. Postulated structures of the studied nanocomposite networks: (a) polyMEA/nano-SiO2; (b) 415 polyMEA/clay/nano-SiO2; (c) polyMEA/clay. It is difficult to understand. Kindly provide what does each structure represent in the scheme.
  12. On page 31, section 4 conclusions, kindly shorten the section and write as conclusion. Following point should be noted while writing this section. (i) Restate the research problem addressed in the paper. (ii)Summarize your overall arguments or findings. (iii)Suggest the key takeaways from your paper.

Author Response

Response to comments of Reviewer #2

The authors would like to thank the Reviewer #2 for his time, his attention, thoughtful comments, and valuable suggestions, which helped to considerably increase the reader-friendliness, as well as the attractivity of the submitted manuscript.

Answers:

Reviewer #2:

The manuscript entitled “Novel tough and transparent ultra-extensible nanocomposite

elastomers based on poly(2-methoxyethylacrylate), and their 3 switching between plasto-elasticity and viscoelasticity” is a document of interesting subject matter particularly studies done on morphology by the SAXS and TEM are very attractive.

However, it needs some major changes before being accepted. Make the following corrections:

Answer: The authors are very grateful for this positive assessment, as well as for the valuable suggestions, which helped to considerably improve the Introduction, as well as the Discussion, and make the Manuscript more reader-friendly and attractive.

The corrections done in response to the comments of Reviewer #2 are highlighted with yellow colour in the copy of the revised Manuscript with highlighted changes.

Reviewer #2:

  1. The interpretation of the experimental results should be significantly improved. In other words, pay attention on the interpretation of the experimental results.

Answer: The Discussion was very thoroughly revised and improved along the suggested lines.

Experimental findings are now described and immediately commented, in a more clear and arranged way. The whole Discussion Part was streamlined and provided with more Headings (for fast and simple orientation in the Discussion), some less important comments were removed.

Important explanative texts (or direct mentions of further-below detailed discussions) which were added in response to this point are highlighted in yellow colour (the numerous stylistic improvements which did not change the content are not highlighted).

Additionally, the Discussion was thoroughly checked for clarity and for reader-friendliness.

Reviewer #2:

  1. The authors discussed the potential of elastomers as efficient polymers for implant technologies, robotics or soft robotics. In this context, other potentials of these nanocomposites, specifically as nano-cement in artificial bones, and scaffolding, should be discussed as well.

Answer:  The original brief mention about application possibilities of the studied nanocomposites was re-written as a new sub-section “1.4 Application potential of the studied materials” in the revised Introduction. Potential applications in artificial ligament, tendon, cartilage, or in nano-cement in artificial bones are now discussed, as well as possible analogous bio-inspired roles in robotic applications, and also the use as scaffold in tissue engineering.

(The re-organization of the revised Introduction is described in the answer to Point 6.).

Reviewer #2:

  1. It is recommended that the table does not have minor horizontal lines. Please check the tables.

Answer: The Table formats were improved as suggested.

Reviewer #2:

  1. Tables must be independent of legible text. Please check tables, especially Table 1, and include the abbreviations (MEA, TMOS, ...) in the caption below of the table.

Answer: The Table Captions were improved as suggested.

Reviewer #2:

  1. In Figure 7, is the results checked statistically? There is a significant difference between analyses? If there is a significant difference statistically, it is recommended to mention it in the caption of the figure.

Answer: These results indeed were checked statistically (3-4 measurements); error bars were added to the revised images of former Figure 7, now sub-images in Figure 6d–f.

Reviewer #2:

  1. I am not convince with the introduction section write up. Kindly rewrite this section again.

Answer: The chapter Introduction was thoroughly re-organized, and is now also structured with the help of some added Headings. The authors are very grateful for the suggestions of Reviewer #2 in points 2., 6., 7., and 9., which helped to make the Introduction much more interesting, and to present the full extent of application potential of the studied materials.

Also the sequence of sub-sections is now smoother and more straight in the revised Introduction:

1.1 Ultra-extensible elastomers and hydrogels

1.2 Nanoparticles and ultra-extensible elastomers

               The architectures of the ultra-extensible (hydro)gels  (= paragraph)

               Inorganic nanofillers  (= paragraph)

               Nano-gels, structured nano-droplets, as well as core-shell nanoparticles 

                                                   (= paragraph)

1.3 Solvent-free hyper-elastomers

               Super-soft solvent-free elastomers  (= paragraph)

               ultra-elastic rubbers   (= paragraph)

1.4 Application potential of the studied materials

1.5 Authors’ previous studies of elastic nanocomposites

1.6 Nanocomposites comparable with the studied ones

               contrast to presented work  (= paragraph)

1.7 Aim of the present work

The reader-friendliness of the revised Introduction was thoroughly checked.

New additions were the paragraph “Nano-gels, structured nano-droplets, as well as core-shell nanoparticles” in the sub-section 1.2, which is dedicated to the particulate nanomaterials and to their application potential,    as well as the much improved sub-section “1.4 Application potential of the studied materials”, which was practically newly written in place of a short paragraph of the original Introduction.

Reviewer #2:

  1. The nanomaterials can be used as efficient materials in cosmetics. In this context, the authors should discuss the development of novel polyMEA nanocomposite elastomers with mentioned new fillers as dermal fillers (lip and facial fillers, or wrinkle fillers). The following papers must be cited and discussed in the introduction: (DOI: 10.1007/s00289-020-03354-6, DOI: 10.3390/polym13071161,)

Answer: The suggested citations about advanced polymer-based nanoparticles able of selective dye absorption, or conversely of drug-release for cancer therapy were added to the thoroughly revised Introduction, to its sub-section "1.2 Nanoparticles and ultra-extensible elastomers", namely to the paragraph about “Nano-gels, structured nano-droplets, as well as core-shell nanoparticles”, where the potential of this type of materials newly is discussed.

In this same paragraph, the use of composite brush-like nanoparticles as dermal fillers in cosmetology and medicine is debated, and also the supramolecular assembly of dermal filler nanoparticles to a viscoelastic material. This latter behaviour is related to the mechanically stronger self-assembly in the studied system, which also is noted in the revised Introduction.

(The re-organization of the revised Introduction is described in the answer to Point 6.).

Reviewer #2:

  1. On page 4 section 2.1. Materials, provide the purity, grade and made of chemicals used in the present work.

Answer: The requested informations about the used chemicals are now provided in the revised Experimental Part.

Reviewer #2:

  1. Kindly include few refrence under introduction section. org/10.3390/polym13183153; doi.org/10.3390/nano11051086

Answer: The suggested citations about drug-releasing nanocomposite micelles, and about self-nanoemulsifying drug delivery systems (SNEDDS) were added to the thoroughly revised Introduction, to its sub-section "1.2 Nanoparticles and ultra-extensible elastomers", namely to the paragraph about “Nano-gels, structured nano-droplets, as well as core-shell nanoparticles”, where the potential of this type of materials newly is discussed.

(The re-organization of the revised Introduction is described in the answer to Point 6.).

Reviewer #2:

  1. On page 11, Figure 3. Morphology of the most important compositions of the studied nanocomposites. Kindly provide the clearly scale bar in each of the TEM images.

Answer: The scalebars in the former Figure 3 (now new Figure 4) were much improved, as suggested.

Reviewer #2:

  1. Scheme 4. Postulated structures of the studied nanocomposite networks: (a) polyMEA/nano-SiO2; (b) 415 polyMEA/clay/nano-SiO2; (c) polyMEA/clay. It is difficult to understand. Kindly provide what does each structure represent in the scheme.

Answer: The schematic representation of the supramolecular structures in the former Scheme 4 (now part of the new Figure 4) was improved by adding explanatory labels.

Reviewer #2:

  1. On page 31, section 4 conclusions, kindly shorten the section and write as conclusion. Following point should be noted while writing this section. (i) Restate the research problem addressed in the paper. (ii)Summarize your overall arguments or findings. (iii)Suggest the key takeaways from your paper.

Answer: The original Conclusions admittedly were not very inviting for the reader. The authors re-wrote this section, in a much simpler and shorter form, and along the lines suggested by the Reviewer. We are very grateful for suggesting this very useful improvement. The section also was renamed, as suggested.

Round 2

Reviewer 1 Report

The authors of this paper have revised the paper draft and the quality of this paper has been improved. The latest draft of this paper reads well. The reviewer suggests publishing this paper. 

Reviewer 2 Report

The authors have satisfactory revised the manuscript.